# Model Shapley: Equitable Model Valuation with Black-box Access

**Xinyi Xu**[†§]**, Thanh Lam**[†]**, Chuan-Sheng Foo**[§]**, Bryan Kian Hsiang Low**[†]
Dept. of Computer Science, National University of Singapore, Republic of Singapore[†]
Inst. for Infocomm Research; Centre for Frontier AI Research, A*STAR, Republic of Singapore[§]
xinyi.xu@u.nus.edu, lamchithanh1997@gmail.com
foo_chuan_sheng@i2r.a-star.edu.sg, lowkh@comp.nus.edu.sg

## Abstract

Valuation methods of data and machine learning (ML) models are essential to the establishment of AI marketplaces. Importantly, certain practical considerations (e.g., operational constraints, legal restrictions) favor the use of model valuation over data valuation. Also, existing marketplaces that involve trading of pre-trained ML models call for an equitable model valuation method to price them. In particular, we investigate the *black-box access* setting which allows querying a model (to observe predictions) without disclosing model-specific information (e.g., architecture and parameters). By exploiting a *Dirichlet abstraction* of a model's predictions, we propose a novel and *equitable* model valuation method called *model Shapley*. We also leverage a Lipschitz continuity of model Shapley to design a learning approach for predicting the model Shapley values (MSVs) of many vendors' models (e.g., $150$) in a large-scale marketplace. We perform extensive empirical validation on the effectiveness of model Shapley using various real-world datasets and heterogeneous model types.

## 1 Introduction

Data valuation methods have important roles in extensive applications such as dataset pricing in a data marketplace [1], evaluating the contributions of data from multiple collaborating parties [71], and identifying valuable data (or filtering out less valuable data) for training higher-quality machine learning (ML) models [20]. However, the following practical considerations favor the use of model valuation over data valuation: (A) Data valuation can be operationally infeasible due to the characteristics of training data, such as being *massively distributed* over millions of sources [73], *enormous in size* (e.g., 400 billion byte-pair-encoded tokens for a language model [7]), and/or *transient* (e.g., the data for online learning are not stored persistently [14]). In contrast, a trained model is *not* distributed by nature (and hence does not need to be aggregated from distributed sources), often much smaller in size than the training data (e.g., less than $1\%$),[1] and usually stored persistently for inference. (B) Data privacy regulations (e.g., GDPR) can be a legal impediment to data valuation in designing *fair* payments for ML-based collaborations (e.g., in medicine [16] or cyber-defense [25]) because the centralization of (possibly) *private* data, which seems necessary for data valuation [39], is prohibited by such regulations. In contrast, the ML models trained on private data without centralization [40] can be available for valuation. (C) Existing marketplaces that involve trading of pre-trained ML models (e.g., AWS marketplace, Modzy) call for an *equitable* model valuation to price them. These practical considerations motivate the use of equitable model valuation over data valuation.

---

[1]The GPT-3 language model contains 175 billion parameters trained on 45 TB of text data [7]. It is of size $1.75 \times 10^{11} \times (2 \times 2) \times 10^{-9} = 350$ GB (at 16 bit float precision) which is less than $1\%$ of training data size.

37th Conference on Neural Information Processing Systems (NeurIPS 2023).

For model valuation, practitioners (e.g., model vendors or clinicians who use the trained models) would likely prefer their models to be examined via only *black-box access*,[2] i.e., by querying the model with input for the corresponding predictions without observing its internal mechanism [8]. It does not disclose the proprietary model information (for model vendors) nor the sensitive information contained in the model (for clinicians), and provides an added advantage of the model valuation being model-agnostic since no model-specific information is used. However, these make model valuation challenging by restricting the available model information to only a selected *query set* (e.g., a validation set) with the observed predictions. In contrast, with white-box access, there is more model information available (e.g., information-theoretic criteria for probabilistic models [71] or norms of the parameters for deep neural networks [22]) to assess model complexity or certain analytical properties (e.g., uniqueness of an optimal model in logistic regression).

Intuitively, the value of a model depends on its intended task: For example, a model trained to classify MNIST digits [45] is not very valuable to a clinician trying to classify diagnostic scans. This dependency is useful in practice for selecting the most 'valuable' model for the desired *task* (i.e., as a query set). We refer to task and query set interchangeably hereafter. While the predictive accuracy of a model on the task provides an intuitive value [66], it can be too reductive: Suppose that two models $\mathbf{M}_i$ and $\mathbf{M}_{i'}$ have identical predictive accuracies on the task but $\mathbf{M}_i$ ($\mathbf{M}_{i'}$) makes highly (barely) certain predictions, i.e., $\mathbf{M}_i$ ($\mathbf{M}_{i'}$) predicts the true class with over $90\%$ (barely over $50\%$) probability. Intuitively, the values for $\mathbf{M}_i$ and $\mathbf{M}_{i'}$ should not be the same, which cannot be achieved with accuracy alone, hence suggesting that additional model information beyond accuracy is required. **(1)** *What then should be a suitable abstraction of a model w.r.t. a task, for model valuation?*

Satisfying certain equitability properties in model valuation is imperative in the application of model pricing to ensuring a fair market. For instance, consistent valuation of identical models is important as it is unfair to price them differently otherwise. Furthermore, the market economy dictates that the value of a model depends on other available models (e.g., more available substitutes cause depreciation), which is important to guarantee a fair market by preventing price fixing (i.e., an exploitative pricing scheme often made illegal by anti-trust laws). **(2)** *How then should model valuation be formulated to satisfy these equitability properties?*

In particular, the Shapley value is shown to satisfy these equitability properties but raises another practical challenge that computing it exactly incurs $\mathcal{O}(2^N)$ time where $N$ is the number of models in a marketplace. So, for a given computational budget, there is a fundamental ceiling to the size of the marketplace such that including more models causes the marketplace to be unable to determine their values (within reasonable time). **(3)** *How can the desirable equitability properties of the Shapley value still be exploited without imposing a significant restriction on the size of the marketplace?*

This paper presents a novel model-agnostic valuation framework to tackle the above challenges. For **(1)**, we use an insight that the predictive pattern of a model (w.r.t. a task) is an suitable abstraction for valuation, especially since the black-box access rules out other model information. Specifically, we use a Dirichlet distribution to approximate a model's predictive pattern/distribution w.r.t. a task, which we call the *Dirichlet abstraction* of this model to encode both its predictive accuracy and certainty (Sec. 2). We describe how to adjust the level of abstraction to trade off the amount of model information (i.e., higher abstraction level) for the availability of a smaller query set. Then, for **(2)**, we observe that ensuring equitability requires a similarity measure of the models (e.g., to ensure identical models are valued consistently or to identify substitutes). We exploit the ability of the Dirichlet abstractions to preserve the similarity between models for proposing *model Shapley* as an equitable model valuation (Sec. 3). As an illustration, identical models produce identical Dirichlet abstractions which result in equal model Shapley values (MSVs). To address **(3)**, based on the Dirichlet abstraction, we leverage a Lipschitz continuity of model Shapley to justify and propose a learning approach of training a *model appraiser* (i.e., a regression learner) on a small subset of models (and their MSVs) for predicting other models' MSVs to validate model Shapley's practical feasibility in a large-scale marketplace (Sec. 4), as empirically verified on real-world datasets and up to $150$ heterogeneous models. We empirically validate that better predictive accuracy (e.g., F1 score) due to better training data, more suitable model types, and/or higher predictive certainty result in higher MSVs, and demonstrate a use case for identifying a valuable subset of models from the marketplace to construct a larger learner (e.g., random forests) (Sec. 5).

---

[2]This is different from black-box models (e.g., deep neural networks) which are difficult to interpret or explain even *with* access to the internal mechanism such as architecture and parameters.

## 2  Dirichlet Abstraction of a Model

We give some preliminaries on the Dirichlet distribution and Hellinger distance below:[3]

**Definition 1** (**Dirichlet distribution** [63]). The probability density function of a $C$-dimensional Dirichlet random variables $Z \sim \mathrm{Dir}(\boldsymbol{\alpha})$ with parameters $\boldsymbol{\alpha} = [\alpha_1, \ldots, \alpha_C] \in (0, \infty)^C$ is $p(z; \boldsymbol{\alpha}) = \prod_{k=1}^{C} z_k^{\alpha_k - 1} \Big/ \left\{ \left[ \prod_{k=1}^{C} \Gamma(\alpha_k) \right] \Big/ \Gamma\left( \sum_{k=1}^{C} \alpha_k \right) \right\}$ where $\Gamma$ is the gamma function.

**Definition 2** (**Hellinger distance** [27]). The Hellinger distance between distributions (whose probability density functions are) $p$ and $q$ is $d_{\mathrm{H}}(p, q) := [1 - \int \sqrt{p(x)\, q(x)}\, \mathrm{d}x]^{1/2}$.

### 2.1  Dirichlet Abstraction and MLE

We will now describe an abstraction of a model via a Dirichlet approximation to the model's predictive pattern over some task and a maximum likelihood estimation (MLE) for it. Each (learnt) $C$-way classification model $\mathbf{M}_i : \mathcal{X} \mapsto \triangle(C)$ is a mapping from the input space $\mathcal{X}$ to the $(C-1)$-probability simplex $\triangle(C)$ (i.e., space of $C$-dimensional probability vectors). Denote a random variable $X \sim P_X$ whose support $\mathrm{supp}(X) = \mathcal{X}$ is the input space, then its distribution $P_X$ induces a predictive distribution $P_{\mathbf{M}_i(X)}$ over $\triangle(C)$. Concretely, $P_X$ is represented by a task/query set $\mathcal{D} := \{(x_j \in \mathcal{X}, y_j \in \{1, \ldots, C\})\}_{j=1,\ldots,D}$ where each $x_j$ is a realization of $X \sim P_X$, so that $\mathbf{M}_i(x_j)$ is a realization of $P_{\mathbf{M}_i(X)}$.

**Dirichlet abstraction.** Note that the predictive distribution $P_{\mathbf{M}_i(X)}$ (induced by $P_X$) (i) has the exactly same support to that of a $C$-dimensional Dirichlet distribution, so $P_{\mathbf{M}_i(X)}$ can be mathematically modeled using a Dirichlet distribution; and (ii) can be statistically modeled using a Dirichlet distribution because the predictive pattern of a classification model can be (statistically) characterized well with a Dirichlet distribution [68]. Informally, the "shape" of predictive distribution of a classification model is similar to that of a Dirichlet distribution. Hence, we let $P_{\mathbf{M}_i(X)} = \mathbb{Q}_i := \mathrm{Dir}(\boldsymbol{\alpha}_i)$, whose parameters $\boldsymbol{\alpha}_i$ can be learnt using MLE (described later) and refer to $\boldsymbol{\alpha}_i$ or $\mathbb{Q}_i$ as $\mathbf{M}_i$'s *Dirichlet abstraction*. In words, we abstract the predictive distribution $P_{\mathbf{M}_i(X)}$ of the model $\mathbf{M}_i$ induced by $P_X$ into a Dirichlet distribution, hence the name Dirichlet abstraction. Note that the notational dependence on $P_X$ (or $\mathcal{D}$) is suppressed when the context is clear.

The proposed Dirichlet abstraction offers several important advantages (further elaborated in Sec. 3): (a) By design, this formulation replaces the heterogeneity in models with the homogeneity in their Dirichlet abstractions, and allows the subsequently proposed model valuation to be applicable to models of different types: The respective Dirichlet abstractions of a multi-layer perceptron and a logistic regression can be compared directly. (b) $\mathbb{Q}_i$ encodes the predictive accuracy and certainty of $\mathbf{M}_i$ through a theoretical connection between the Hellinger distance $d_{\mathrm{H}}$ of two Dirichlet abstractions and the cross entropy of $\mathbf{M}_i$ (formalized by Proposition 2). (c) Importantly for model valuation, an appealing analytic property of the Dirichlet distribution is that $d_{\mathrm{H}}$ of two Dirichlet abstractions can be evaluated in closed-form and incurs $\mathcal{O}(1)$ computational complexity.

**MLE.** We adopt the MLE approximation of $\boldsymbol{\alpha}_i$ using the predictions of $\mathbf{M}_i$ on $\mathcal{D}$ since the log-likelihood function is concave with a unique maximizer and can thus be efficiently optimized [57]. Since $\mathbf{M}_i(x_j)$ denotes a realized predictive probability vector of $\mathbf{M}_i(X) \sim \mathrm{Dir}(\boldsymbol{\alpha}_i)$, we use the *observed sufficient statistics* $\log \bar{h}_i := D^{-1} \sum_{j=1}^{D} \log \mathbf{M}_i(x_j)$ (i.e., with an element-wise log operation) to derive the log-likelihood as follows [36]:

$$F(\boldsymbol{\alpha}, \bar{h}_i) := D \left[ G(\boldsymbol{\alpha}) + \sum_{k=1}^{C} (\alpha_k - 1) \log \bar{h}_{i,k} \right] \tag{1}$$

where $G(\boldsymbol{\alpha}) := \log \Gamma\left( \sum_k \alpha_k \right) - \sum_k \log \Gamma(\alpha_k)$. From (1), $\bar{h}_i$ arises as an alternative (to $\boldsymbol{\alpha}_i$) abstraction of $\mathbf{M}_i$ w.r.t. $\mathcal{D}$. Hence, we compare $\boldsymbol{\alpha}_i$ and $\bar{h}_i$ theoretically (Proposition 3 in App. A) and empirically (Sec. 4.2).

---

[3]Additional discussion on the choice (e.g., a comparison with the Chernoff distance) and suitability of Hellinger distance is provided in App. A.

## 2.2 Class-specific Dirichlet Abstraction

Since the Dirichlet abstraction $\mathbb{Q}_i$ of $\mathbf{M}_i$ w.r.t. some task $\mathcal{D}$ does *not* explicitly account for the class information in $\mathcal{D}$ (i.e., the true classification label $y_j$ of each data $x_j$), models with certain (different) predictive patterns can be indistinguishable. This can be problematic because the Dirichlet abstractions of an optimal model and another model with zero predictive accuracy but a specific predictive pattern can identical, making it difficult to distinguish between these two models.

Suppose that there are $C = 3$ classes in a balanced query set $\mathcal{D}$ (i.e., equal data size for each class). For some $\mathbf{M}_i$ (with optimal predictive accuracy), artificially construct $\mathbf{M}_{i'}$ to have the same Dirichlet abstraction, but *zero* predictive accuracy in the following way: Since $\mathcal{D}$ is balanced, group the data into triplets $\{(x_{j,1}, x_{j,2}, x_{j,3})\}_{j=1,\dots,D/3}$ for the input data from 3 classes. Next, define $\mathbf{M}_{i'}(x_{j,c}) := \mathbf{M}_i(x_{j,(c \bmod 3)+1})$ for $c = 1, 2, 3$. Intuitively, for each prediction (i.e., a 3-dimensional probability vector in a 2-simplex) by $\mathbf{M}_i$, $\mathbf{M}_{i'}$ makes an identical one but on an input data from a wrong class, i.e., we 'shift' $\mathbf{M}_i$'s predictions by one class to construct the predictions of $\mathbf{M}_{i'}$. Obviously, the predictive accuracy of $\mathbf{M}_{i'}$ is zero, but its predictions (in aggregation) are identical to those of $\mathbf{M}_i$, which means $\bar{h}_i = \bar{h}_{i'}$ in (1), hence resulting in $\mathbb{Q}_i = \mathbb{Q}_{i'}$. In Fig. 1: Plots 1 and 2 are the predictions of $\mathbf{M}_i$ and $\mathbf{M}_{i'}$ respectively, on $\mathcal{D}$, which are (visually) indistinguishable. Plots 3 and 4 are samples from $\mathbb{Q}_i$ and $\mathbb{Q}_{i'}$ respectively, which are also (visually) indistinguishable. The implication is that, in this case (of $\mathbf{M}_i$ and such specially constructed $\mathbf{M}_{i'}$), the highest level of abstraction (i.e., using the entire query set $\mathcal{D}$ to construct Dirichlet abstractions $\mathbb{Q}_i$ and $\mathbb{Q}_{i'}$) is not effective to distinguish between $i$ and $i'$. Hence, we adopt a lower level of abstraction, described next.

The remedy is the so-called *class-specific Dirichlet abstraction*: Partition the query set $\mathcal{D} = \cup_{k=1}^C \mathcal{D}_k$ where $\mathcal{D}_k$ contains *only* data from the $k$-th class. The Dirichlet abstraction $\boldsymbol{\alpha}_{i,\mathcal{D}_k}$ of $\mathbf{M}_i$ on $\mathcal{D}_k$ is called the class-specific Dirichlet abstraction w.r.t. class $k$. Based on the example above, we verify using a small experiment.[4] Over the entire $\mathcal{D}$, we obtain $\boldsymbol{\alpha}_i = \boldsymbol{\alpha}_{i'} = [0.5040, 0.5339, 0.5306]$. Restricting to query set $\mathcal{D}_1$ from class 1 only, gives $\boldsymbol{\alpha}_{i,\mathcal{D}_1} = [21.8601, 2.2005, 2.0215]$ and $\boldsymbol{\alpha}_{i',\mathcal{D}_1} = [1.5817, 1.7576, 19.4037]$, which are clearly different. In Fig. 1: Plots 5 and 6 are the predictions of $\mathbf{M}_i$ and $\mathbf{M}_{i'}$, respectively, on $\mathcal{D}_1$ while plots 7 and 8 are samples from $\mathbb{Q}_{i,\mathcal{D}_1}$ and $\mathbb{Q}_{i',\mathcal{D}_1}$. We see that $\mathbf{M}_i$ is clearly different from $\mathbf{M}_{i'}$, and $\mathbb{Q}_{i,\mathcal{D}_1}$ is clearly different from $\mathbb{Q}_{i',\mathcal{D}_1}$, demonstrating the effectiveness of a lower level of abstraction via the class-specific Dirichlet abstraction.

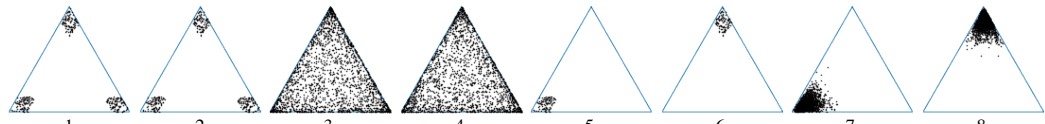

Figure 1: The triangles denote the 2-simplex and each dot is a 3-dimensional probability vector. Plots 1-4 (5-8) use $\mathcal{D}$ ($\mathcal{D}_1$) as query set. Plots 1&5 (2&6) show predictions of $\mathbf{M}_i$ ($\mathbf{M}_{i'}$). Plots 3&7 (4&8) show randomly drawn samples from Dirichlet abstraction $\mathbb{Q}_i$ ($\mathbb{Q}_{i'}$). For example, plot 4 shows random samples from $\mathbb{Q}_{i'}$ w.r.t. $\mathcal{D}$, while plot 5 shows predictions of $\mathbf{M}_i$ w.r.t. $\mathcal{D}_1$.

**Trade-off between level of Dirichlet abstraction and size of query set.**  Note that we need not stop at partitioning $\mathcal{D}$ at the class level. For instance, a more refined partition can be first according to the classes and then certain groups of input feature values. Doing so produces Dirichlet abstractions with a *lower* level of abstraction (i.e., less abstract and containing more model information), but also requires a *larger* query set $\mathcal{D}$ to begin with so that each smaller partitioned query set contains sufficient data (for predictions) to obtain an accurate MLE estimate.[5] For an extremely refined partition of $\mathcal{D}$ where each partitioned query set is very small (e.g., size $\leq 5$), the obtained MLE estimates and the corresponding Dirichlet abstraction can be inaccurate and thus not useful. It is thus important to find a suitable trade-off between the abstraction level vs. query set size. In particular, in Sec. 5, we observe that partitioning $\mathcal{D}$ according to the classes provides such a suitable trade-off: E.g., it can distinguish models with almost equal predictive accuracy due to the high class imbalance

---

[4]On a balanced query set of size $D = 900$, $\mathbf{M}_i(x_j) := G_j[\mathbb{1}(y_j = c) + \epsilon_c]_{c=1,2,3}$ with $\epsilon_c \sim \mathcal{U}(0, 0.2)$ and normalizing constant $G_j$, i.e., $\mathbf{M}_i$ always makes the correct classification but with a small additive and independent uniform noise $\epsilon_c$.

[5]The work of [3] shows a sample complexity of $|\mathcal{D}|$ being polynomial in $C$ and an additive error.

in the data[6] but different F1 scores. Since the Dirichlet abstraction and the class-specific version are both Dirichlet distributions, and share the same theoretical properties (e.g., enabling a closed-form expression of the Hellinger distance), subsequently, we refer to Dirichlet abstractions generally (i.e., omitting the specific dependence on $\mathcal{D}$ or $\mathcal{D}_k$), unless otherwise specified.

## 3 Equitable Valuation via Model Shapley

We discuss and formalize several equitability properties to derive a general formulation called *model Shapley* to satisfy them. Then, we will specify the *characteristic function* and a *precision-weighted fusion* to encode a model's predictive accuracy and certainty into its MSV.

### 3.1 Equitability Properties and Model Shapley

Consider $N$ models in a marketplace and denote $\mathbf{M}_i$'s equitable value by $\phi_i$. (P1) If a model $\mathbf{M}_i$ has not been queried at all, then its value is indeterminate and we set $\phi_i = 0$ by default. (P2) If two models $\mathbf{M}_i$ and $\mathbf{M}_{i'}$ give identical predictions (over the task), then their values are equal, i.e., $\phi_i = \phi_{i'}$. (P3) If some buyer is interested in multiple tasks simultaneously and a model (from some vendor) performs very well on *only* one of the tasks, then it is unfair for the vendor to set the value/price solely based on this performance. Instead, an equitable value should be based on the model's joint performance on these tasks (e.g., a linear combination according to the buyer's interests in these tasks). (P4) The existence of perfect substitutes depreciates the value of a model. (P1)-(P4) are useful for equitable valuation in ML model marketplaces [51, 66]. To formalize these properties, some notations are needed: Let $\nu : 2^{[N]} \mapsto \mathbb{R}$ denote a characteristic function (specified later) s.t. $\nu(\mathcal{C})$ quantifies the value of a collection $\mathcal{C} \subseteq [N] := \{1, \ldots, N\}$ of models to capture the dependence of $\mathbf{M}_i$'s value on other existing models $\mathbf{M}_{i'}$ (e.g., substitutes). Denote $\mathbf{M}_i$'s value by $\phi_i \leftarrow \Phi(i, \nu, \{\mathbf{M}_i\}_{i \in [N]})$ which is fully specified (up to linear scaling) by the properties:

P1 Null player: $(\forall \mathcal{C} \subseteq [N] \setminus \{i\} \ \nu(\mathcal{C} \cup \{i\}) - \nu(\mathcal{C}) = 0) \implies \phi_i = 0$.
P2 Symmetry: $(\forall \mathcal{C} \subseteq [N] \setminus \{i, i'\} \ \nu(\mathcal{C} \cup \{i\}) = \nu(\mathcal{C} \cup \{i'\})) \implies \phi_i = \phi_{i'}$.
P3 Linearity: $\forall \gamma, \gamma' \in \mathbb{R} \ (\nu_{\mathcal{D} \cup \mathcal{D}'} := \gamma \, \nu_{\mathcal{D}} + \gamma' \, \nu_{\mathcal{D}'}) \implies \phi_i(\mathcal{D} \cup \mathcal{D}') = \gamma \, \phi_i(\mathcal{D}) + \gamma' \, \phi_i(\mathcal{D}')$.
P4 Diminished marginal utility: Add a perfect substitute (i.e., duplicate/copy) $\mathbf{M}_{i_c}$ of $\mathbf{M}_i$ to the pool of $N$ models already containing $\mathbf{M}_i$ and denote the new pool by $[N'] := [N] \cup \{i_c\}$. Denote the value of $\mathbf{M}_i$ w.r.t. $[N]$ by $\phi_i$ and w.r.t. $[N']$ by $\phi_i'$. Then, $\phi_i' \leq \phi_i$.

**Proposition 1.** Properties P1, P2, and P3 fully specify $\Phi(\cdot)$ up to a linear scaling $Z$:

$$\phi_i := Z \sum_{\mathcal{C} \subseteq [N] \setminus \{i\}} \omega_{\mathcal{C}} \left[ \nu(\mathcal{C} \cup \{i\}) - \nu(\mathcal{C}) \right] \quad \text{where } \omega_{\mathcal{C}} := |\mathcal{C}|! \times (N - |\mathcal{C}| - 1)!/N! . \quad (2)$$

Its proof follows directly from [20, Proposition 2.1]. Since (2) coincides with the Shapley value [69], we refer to $\Phi$ as model Shapley and $\phi_i$ as the *model Shapley value* (MSV). Note that P3 requires two distinct tasks $\mathcal{D}$ and $\mathcal{D}'$ to distinguish between the MSVs of $\mathbf{M}_i$ w.r.t. to these two tasks. P4 additionally requires $\nu$ to be *conditionally redundant*[7] (Proposition 5 in App. A): The benefit of a redundant copy $\mathbf{M}_{i_c}$ (conditioned on model $\mathbf{M}_i$ already being added) is not more than the initial benefit of adding $\mathbf{M}_i$. Intuitively, as adding $\mathbf{M}_i$ is already sufficient for the desired task, subsequently adding $\mathbf{M}_{i_c}$ does not yield (as much) extra benefit [24]. In contrast, [51] also adopts (2) but assumes $\nu$ already exists while we explicitly design $\nu$ to encode predictive accuracy and certainty:

$$\nu(\mathcal{C}) := -d(\mathbb{Q}_{\mathcal{C}}, \mathbb{Q}^*) \tag{3}$$

where $\mathbb{Q}_{\mathcal{C}}$ is the *precision-weighted fusion* of the Dirichlet abstractions in $\mathcal{C}$ and $d(\cdot, \cdot)$ is a distributional distance measure between two Dirichlet abstractions. Recall from Sec. 2 that the predictive distribution of $\mathbf{M}_i$ is represented in its Dirichlet abstraction (visualized in Fig. 1). In particular, the more accurate $\mathbf{M}_i$ is, the closer (distributionally) $\mathbb{Q}_i$ is to the Dirichlet abstraction $\mathbb{Q}^*$ of an expert (i.e., optimal classifier). Hence, a (high) similarity between $\mathbb{Q}_i$ and $\mathbb{Q}^*$ can suggest $\mathbf{M}_i$'s (high) predictive accuracy. Specifically, $\mathbb{Q}^*$ is implemented as follows: For an input data-label pair $(x_j, y_j)$, take the one-hot encoded vector $e_{y_j} \in [0, 1]^C$ of $y_j$, add an independent uniform noise $\epsilon_j \in [0, 0.01]^C$ (to avoid degeneracy during MLE), normalize it to sum to 1 to yield $e_{y_j, \epsilon_j}$ as a

---

[6]98.22% of the data are from only 3 out of the 23 classes.
[7]$\forall \mathcal{C} \subseteq [N] \setminus \{i\} \ \nu(\mathcal{C} \cup \{i\}) - \nu(\mathcal{C}) \geq \nu(\mathcal{C} \cup \{i, i_c\}) - \nu(\mathcal{C} \cup \{i\})$ given that $i_c \notin [N]$.

'prediction' of the expert, and solve (1) using these 'predictions' over $\mathcal{D}$. The predictive certainty of $\mathbf{M}_i$ is encoded in the precision $|\boldsymbol{\alpha}_i|_1$ of its Dirichlet abstraction, which is then used as a weight to fuse the individual Dirichlet abstractions $\mathbb{Q}_i$ in $\mathcal{C}$ to obtain $\mathbb{Q}_{\mathcal{C}}$:

**Definition 3** (**Precision-weighted Fusion**). Let the random vector $[\beta_1, \ldots, \beta_n] \sim$ $\mathrm{Dir}([|\boldsymbol{\alpha}_1|_1, \ldots, |\boldsymbol{\alpha}_n|_1])$ be independent of $\mathbf{M}_i(x)$ for all $i \in \mathcal{C} \subseteq [N]$ where $n := |\mathcal{C}|$. Then, the precision-weighted fusion $\mathbb{Q}_{\mathcal{C}}$ is the distribution of $\sum_{i=1}^{n} \beta_i \, \mathbf{M}_i(x)$.

In Definition 3, the weight $\beta_i$ on $\mathbf{M}_i(x)$ is large if $|\boldsymbol{\alpha}_i|_1$ is large, i.e., $\mathbb{Q}_i$ has a high precision/$\mathbf{M}_i$ has a high predictive certainty. Definition 3 has an important implication: $\mathbb{Q}_{\mathcal{C}}$ is a fully specified Dirichlet, i.e., $\mathbb{Q}_{\mathcal{C}} = \mathrm{Dir}([\sum_{i=1}^{n} \alpha_{i,1}, \ldots, \sum_{i=1}^{n} \alpha_{i,C}])$ (Lemma 3 in App. A) with four advantages: (A) $\mathbb{Q}_i$ and $\mathbb{Q}_{\mathcal{C}}$ are both Dirichlets so that a single $d$ (e.g., $d_{\mathrm{H}}$) in (3) can be used for both singleton and non-singleton $\mathcal{C}$'s. Interestingly, it gives a perspective that each $\mathbf{M}_i$ lives as $\mathbb{Q}_i$ in a metric space w.r.t. $d_{\mathrm{H}}$ (Fig. 11 in App. C). (B) We can theoretically justify learning model Shapley (Theorem 1). (C) (3) using $d_{\mathrm{H}}$ can be evaluated in closed form with $\mathcal{O}(1)$ time, which is important since (2) requires $\mathcal{O}(2^N)$ evaluations of $\nu(\mathcal{C})$ for different $\mathcal{C}$'s. (D) An alternative to specify $\nu(\mathcal{C})$ is a linear combination of the performance of $\mathbf{M}_i, \forall i \in \mathcal{C}$ as $\nu(\mathcal{C})$. However, it is unclear what the weights in this linear combination should be. In contrast, Definition 3 'automatically' resolves this issue by fusing each $\mathbb{Q}_i$ according to its precision $|\boldsymbol{\alpha}_i|_1$ into $\mathbb{Q}_{\mathcal{C}}$.

**Interpreting MSV.** Note that (2) is an average (over all possible $\mathcal{C} \subseteq [N] \setminus \{i\}$) of how much $\mathbf{M}_i$ (or, more precisely, $\mathbb{Q}_i$) improves the distributional similarity between $\mathbb{Q}_{\mathcal{C}}$ and $\mathbb{Q}^*$ (i.e., the expert) after $i$ joins $\mathcal{C}$. Both the predictive accuracy and certainty of $\mathbf{M}_i$ can affect (2). To see this, a high predictive accuracy of $\mathbf{M}_i$ implies that $\mathbb{Q}_i$ is (distributionally) close to $\mathbb{Q}^*$; a high predictive certainty of $\mathbf{M}_i$ ensures its weight $\beta_i$ is large when fused into $\mathbb{Q}_{\mathcal{C}}$, so the predictive certainty can amplify $\mathbf{M}_i$'s effect in bringing $\mathbb{Q}_{\mathcal{C}}$ close to $\mathbb{Q}^*$ (if $\mathbb{Q}_i$ is close to $\mathbb{Q}^*$). *Hence, a model $\mathbf{M}_i$ with a high predictive accuracy and certainty is likely to have a high MSV $\phi_i$.* When considering several models jointly [66], we can use $\phi_i$ to indicate how well (on average) $\mathbf{M}_i$ combines with other models (i.e., whether $\mathbf{M}_i$ joining $\mathcal{C}$ leads to a performance improvement), as verified in Sec. 5. In contrast to the work of [66], which supports up to $N = 32$ *simpler* binary classifiers, our approach –more scalable and general– supports up to $N = 150$ $C$-way classifiers (Sec. 4.2).

## 3.2 Connection to Cross-entropy and Other Model Evaluation Criteria

We first formalize the connection between cross-entropy (CE) and our approach that uses the Hellinger distance, and then use this connection to extend our approach to other evaluation criteria such as adversarial robustness [43], distributional robustness [67], and algorithmic fairness in ML [56].

As a distributional distance measure, CE can be used specify (3) because the CE loss is used to evaluate the performance of a model. Then, specifying (3) with CE or the Hellinger distance (as proposed in Sec. 4) can be connected in how they evaluate the model performance.

**Proposition 2.** Let $\nu_{\mathrm{CE}}(\mathcal{C}) := -\mathrm{CE}(\mathbb{Q}_{\mathcal{C}}, \mathbb{Q}^*),$[8] $\nu_{\mathrm{H}}^2(\mathcal{C}) := -d_{\mathrm{H}}^2(\mathbb{Q}_{\mathcal{C}}, \mathbb{Q}^*)$ and $H(\cdot)$ be differential entropy. Then,

$$\nu_{\mathrm{CE}}(\mathcal{C}) \leq \nu_{\mathrm{H}}^2(\mathcal{C}) \text{ if } H(\mathbb{Q}_{\mathcal{C}}) + d_{\mathrm{H}}^2(\mathbb{Q}_{\mathcal{C}}, \mathbb{Q}^*) \geq 0 \text{, and } \nu_{\mathrm{CE}}(\mathcal{C}) \geq \mathrm{const}_{\mathbb{Q}^*} \times ([1 + \nu_{\mathrm{H}}^2(\mathcal{C})]^2 - 1) + \log \Gamma(C)$$

where $\mathrm{const}_q := [\log(1/q_{\min} - 1)]/(1 - 2q_{\min})$ with $q_{\min} := \min_z q(z)$ for a density $q$ .

Proposition 2 shows that $\nu_{\mathrm{CE}}(\mathcal{C})$ has upper and lower bounds that are monotonic in $\nu_{\mathrm{H}}^2$, providing some justification for using $\nu_{\mathrm{H}}$ (instead of $\nu_{\mathrm{CE}}$). Note that while Proposition 2 utilizes $\nu_{\mathrm{H}}^2$ due to a key Lemma 2 (in App. A), we adopt $\nu_{\mathrm{H}} := -d_{\mathrm{H}}$ (Sec. 4) as $d_{\mathrm{H}}$ satisfies the triangle inequality (for Theorem 1). Moreover, Proposition 2 confirms that $\nu_{\mathrm{H}}$ encodes the predictive accuracy and certainty of a model since $\nu_{\mathrm{CE}}$ encodes the predictive accuracy and certainty (see the example in App. A).

Interestingly, this connection to CE enables the extension to adversarial robustness, distributional robustness and algorithmic fairness, with different practical motivations. For instance, adversarial robustness (in model valuation) is important to application scenarios where the model can encounter adversarial attacks in deployment [43]. Formally, the respective objective functions objective$_{\mathrm{adv}}$ [43], objective$_{\mathrm{DRO}}$ [67, Equation 5] and objective$_{\mathrm{EO}}$ [56, Definition 2] can be achieved from a suitable definition of $\nu$ using $d_{\mathrm{H}}$ (precise definitions and full deviations are deferred to App. A), as summarized

---

[8]Note that $\mathrm{CE}(p, q) := -\int p(x) \log q(x) \, \mathrm{d}x$ for two distributions with densities $p, q$.

in Table 1. We highlight that this illustrates the potential generality of MSVs using the Hellinger distance w.r.t. Dirichlet abstractions (i.e., using $\nu_{\mathrm{H}}(\mathcal{C})$ in (2)), and defer the formal treatment of such theoretical connections to future work. A question one might ask is that: (How) can multiple such evaluation criteria be combined? The answer is yes, by leveraging (P3) to *linearly combine* selected evaluation criteria, as discussed in App. A.

Table 1: Extension to other evaluation criteria for model valuation. The notation dependence on the query set is made explicit. $\mathcal{D}_{\mathrm{adv}}, \mathcal{D}_{\mathrm{clean}}$ denote the query sets containing adversarial and non-adversarial (clean) training examples. $\{\mathcal{D}_g\}_{g \in \mathcal{G}}$ is a collection of query sets where $\mathcal{D}_g$ contains training examples from a particular "group"/data distribution. $\mathcal{D}_{\mathrm{prot}}^+, \mathcal{D}_{\mathrm{unprot}}^+$ contain positive training examples under the protected and unprotected groups, respectively.

| Criteria | Query sets | Choices of $\nu$ |
|---|---|---|
| objective$_{\mathrm{adv}}$ | $\mathcal{D}_{\mathrm{adv}}, \mathcal{D}_{\mathrm{clean}}$ | $-(d_{\mathrm{H}}(\mathbb{Q}, \mathbb{Q}^*; \mathcal{D}_{\mathrm{adv}}) + \upsilon \, d_{\mathrm{H}}(\mathbb{Q}, \mathbb{Q}^*; \mathcal{D}_{\mathrm{clean}}))$ |
| objective$_{\mathrm{DRO}}$ | $\{\mathcal{D}_g\}_{g \in \mathcal{G}}$ | $-\max_{g \in \mathcal{G}} d_{\mathrm{H}}(\mathbb{Q}, \mathbb{Q}^*; \mathcal{D}_g)$ |
| objective$_{\mathrm{EO}}$ | $\mathcal{D}_{\mathrm{prot}}^+, \mathcal{D}_{\mathrm{unprot}}^+$ | $-|d_{\mathrm{H}}(\mathbb{Q}, \mathbb{Q}^*; \mathcal{D}_{\mathrm{prot}}^+) - d_{\mathrm{H}}(\mathbb{Q}, \mathbb{Q}^*; \mathcal{D}_{\mathrm{unprot}}^+)|$ |

# 4 Learning Model Shapley

To address challenge **(3)** in Sec. 1, we propose a learning approach to train a model appraiser (i.e., a regression learner) from the MSVs of a small subset of models for predicting MSVs of the remaining models (further elaborated in App. C). If we can learn a good appraiser with only $20\%$ of all models (empirically verified), then the marketplace size can (theoretically) quintuple. To justify this learning approach, we derive a Lipschitz continuity of model Shapley, which is also empirically verified using 5 real-world datasets and various model types. Next, we implement this learning approach by training a Gaussian process regression (as the model appraiser) on a subset (from $5\%$ to $50\%$ in size) of 150 model-MSV pairs and examine its predictive performance on the rest. Our implementation is available at `https://github.com/XinyiYS/ModelShapley`.

## 4.1 Lipschitz Continuity of Model Shapley

We derive a Lipschitz continuity of the model Shapley function $\Phi : [N] \mapsto \mathbb{R}$: The difference between the MSVs of two models $\mathbf{M}_i, \mathbf{M}_{i'}$ (i.e., inputs to $\Phi$) is bounded by the distance $d_{\mathrm{H}}(\mathbb{Q}_i, \mathbb{Q}_{i'})$ between them, multiplied by a constant factor.

**Theorem 1 (Lipschitz Continuity).** Let $d := d_{\mathrm{H}}$ in (3). Then, $\forall i, i' \in [N] \ (\forall \mathcal{C} \subseteq [N] \setminus \{i, i'\} \ \ d_{\mathrm{H}}(\mathbb{Q}_{\mathcal{C} \cup i}, \mathbb{Q}_{\mathcal{C} \cup i'}) \leq d_{\mathrm{H}}(\mathbb{Q}_i, \mathbb{Q}_{i'})) \implies |\phi_i - \phi_{i'}| \leq Z d_{\mathrm{H}}(\mathbb{Q}_i, \mathbb{Q}_{i'}).$

Its proof is in App. A. Theorem 1 states that the difference in MSVs of two models is bounded by the Hellinger distance between their Dirichlet abstractions, and the constant $Z$ from (2) is the Lipschitz constant. This is based on a simple fusion-increases-similarity condition: When $\mathbb{Q}_i$ and $\mathbb{Q}_{i'}$ are each fused with a common $\mathbb{Q}_{\mathcal{C}}$, the resulting similarity is higher (i.e., smaller $d_{\mathrm{H}}$ since $\mathbb{Q}_{\mathcal{C} \cup i}$ and $\mathbb{Q}_{\mathcal{C} \cup i'}$ have $\mathbb{Q}_{\mathcal{C}}$ in common (see Proposition 4 in App. A). Moreover, Table 3 and Fig. 6 (in App. A) empirically verify Theorem 1. Then, Theorem 1 provides a theoretical justification for the learning approach because it guarantees that similar inputs (i.e., small $d_{\mathrm{H}}(\mathbb{Q}_i, \mathbb{Q}_{i'})$) imply similar outputs (i.e., small $|\phi_i - \phi_{i'}|$). namely, the model Shapley function is well-behaved w.r.t. its inputs, and hence learnable. This reasoning is applied to justify learning the data Shapley value [21].

## 4.2 Empirical Learning Performance via Gaussian Process Regression (GPR)

To exploit the Lipschitz continuity (i.e., Theorem 1), we adopt the Gaussian process regression (GPR) due to a uniform error bound of GPR on Lipschitz continuous functions [47]. Our implementation trains a GPR (as the model appraiser) on the MSVs of a subset of $N = 150$ models and examines its predictive performance on the remaining ones.

**Regression setting.** We train $N = 150$ independent models on MNIST (CIFAR-10): 50 of logistic regression (LR), multi-layer perceptron (MLP), and convolutional neural network (CNN) each (ResNet-18, SqueezeNet, and DenseNet-121 each). For simplicity, we use the test set *without*

partitioning as the query set $\mathcal{D}$. For each $\mathbf{M}_i$, we obtain $\bar{h}_i$ via its predictions on $\mathcal{D}$ and solve (1) to obtain $\boldsymbol{\alpha}_i$ as input features for separate regressions. For the regression labels, as calculating $\phi_i$ exactly incurs $\mathcal{O}(2^{150})$ time, we use the $(\epsilon = 0.1, \delta = 0.1)$-approximation [53] $\hat{\phi}_i$ as the average of 3745 Monte-Carlo samples. This results in two sets $\{\boldsymbol{\alpha}_i, \hat{\phi}_i\}$ and $\{\bar{h}_i, \hat{\phi}_i\}$ of model-MSV pairs of size 150 each. We train a GPR on a random subset of 150 model-MSV pairs to learn to predict the MSV on the remaining pairs. In GPR, we use the squared exponential kernel $\exp(-d(i,i')/(2\sigma^2))$ (the lengthscale $\sigma$ is learnt) where $d(i,i') := d_{\mathrm{H}}(\mathbb{Q}_i, \mathbb{Q}_{i'})$ for $\{\boldsymbol{\alpha}_i, \hat{\phi}_i\}$, and $d(i,i') := |\bar{h}_{i'} - \bar{h}_i|_1$ for $\{\bar{h}_i, \hat{\phi}_i\}$ .

**High regression performance verifies learnability.** We examine the test performance using two error metrics: mean-squared error (MSE) and maximum error (MaxE) w.r.t. varied training ratios from 5% to 50%, in Fig. 2. In particular, results for training ratio of 20% are in Table 2. We observe that even using only 20% of model-MSV pairs for training, the learning is effective (i.e., low test errors), which shows its feasibility in a large-scale model marketplace. This can be attributed to the learnability justified by Theorem 1 and the uniform error bound of GPR [47]. In addition, learning on $\boldsymbol{\alpha}_i$ is more effective than learning on $\bar{h}_i$ (as Table 2 and Fig. 2 show higher errors for the latter), since the average operation to get $\bar{h}_i$ loses some model information.

Table 2: Top (bottom) are results on MNIST (CIFAR-10) for the training ratio of 20%. Average (std. error) over 10 random train-test splits.

|  | MSE | MaxE |
|---|---|---|
| $\boldsymbol{\alpha}_i$ | $\mathbf{1.59e^{-6}}(6.9e^{-8})$ | $\mathbf{3.53e^{-3}}(9.7e^{-5})$ |
| $\bar{h}_i$ | $8.36e^{-5}(3.1e^{-6})$ | $1.59e^{-2}(2.6e^{-4})$ |
| $\boldsymbol{\alpha}_i$ | $\mathbf{1.79e^{-5}}(5.2e^{-6})$ | $\mathbf{9.05e^{-3}}(3.9e^{-4})$ |
| $\bar{h}_i$ | $3.05e^{-4}(4.5e^{-5})$ | $3.23e^{-2}(2.4e^{-3})$ |

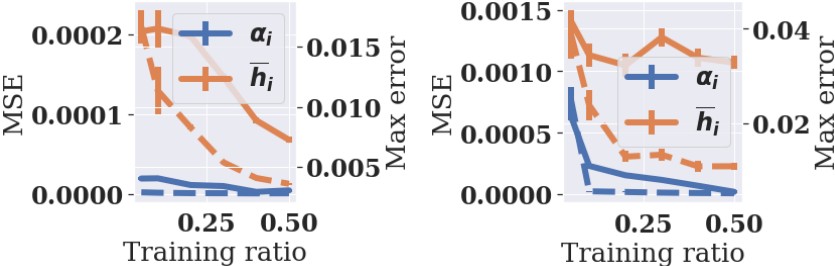

Figure 2: Average (std. errors) of test performance vs. training ratios for MNIST & CIFAR-10 over 10 random train-test splits for each training ratio. Dashed (solid) lines follow the left (right) axis. Colors indicate $\boldsymbol{\alpha}_i$ (blue) or $\bar{h}_i$ (orange). To elaborate, at a training ratio of 5%, the GPR is trained on a random subset of 5% of the total 150 model-MSV pairs and its test performance on the remaining 95% of the model-MSV pairs is reported.

# 5 MSV vs. Common Evaluation Criteria

This work is motivated by the lack of a standardized model valuation, but there are different evaluation criteria useful in different scenarios (e.g., accuracy, F1 score, predictive certainty). Interestingly, we find that MSV can produce consistent model values with these criteria. Additionally, we evaluate the utility of MSV directly in a use case where a buyer wishes to purchase multiple models with black-box access to build a larger learner (e.g., random forest or voting classifier) [33, 66] and show that MSV can be used to effectively identify the most 'valuable' models for this purpose.

**MSV vs. predictive performance.** Fig. 3 (left) compares the MSVs of different model types (independently trained on the same data) for MNIST. Here $\mathcal{D}$ consists of misclassified data (from the original test set) of all the models to highlight the difference in their predictive accuracies. Without

needing to partition $\mathcal{D}$, we observe that CNNs significantly outperform both MLPs and LRs (in terms of accuracy) and have the highest MSVs. This is expected, since CNNs are more capable of performing well in image-based tasks, and hence the MSVs for CNNs are correspondingly higher. Then, we examine predictive certainty. We use the same CNN model type independently trained on the same MNIST data (i.e., their accuracies are essentially equal), but artificially increase the predictive certainty for some: We multiply the highest probability of $\mathbf{M}_i(x_j)$ by a factor of $[1, 5, 10]$ and then normalize the resulting vector to sum to 1 *without* affecting the predicted class/accuracy. Fig. 3 (right) shows that models with higher predictive certainty have higher MSVs, confirmed by additional results on CIFAR-10, MedNIST, and DrugRe in App. C. These results confirm our intuition that a model with high predictive accuracy and certainty is likely to have a high MSV.

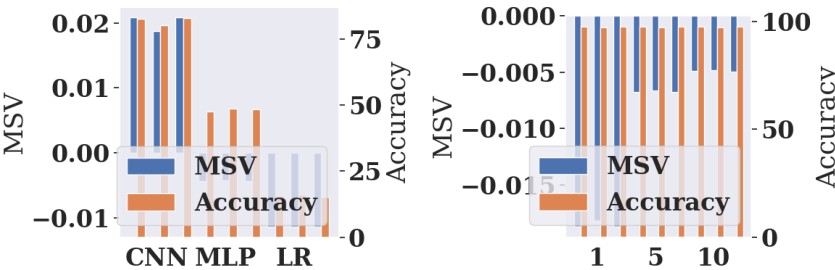

Figure 3: More suitable model types (left)/higher predictive certainty (right) lead to higher MSVs.

Next, we train 3 groups of 3 LR models independently with varying sizes of training data from $0.001$ to $0.01$ and $0.1$ of the entire KDD99 dataset containing highly imbalanced data; the top 3 classes out of 23 constitute $98.22\%$ of total data. Intuitively, the models trained on more data should perform better and thus be more valuable, but due to the class imbalance, Fig. 4 (left) shows difficulties in differentiating these models based on their accuracies (or MSVs w.r.t. the entire/unpartitioned query set). Intuitively, to differentiate them, we need a lower level (i.e., more refined) Dirichlet abstraction, namely the class-specific Dirichlet abstractions: Partition the entire query set according to the $C = 23$ classes with $\gamma_k := |\mathcal{D}_k|$. Define $d_{\mathrm{H}}(\mathbb{Q}_i, \mathbb{Q}_{i'}; \{\mathcal{D}_k\}_{k=1,\ldots,C}) := \sum_{k=1}^{C} \gamma_k \, d_{\mathrm{H}}(\mathbb{Q}_{i,\mathcal{D}_k}, \mathbb{Q}_{i',\mathcal{D}_k})$ to leverage P3 to compute $\phi_i(\{\mathcal{D}_k\}_{k=1,\ldots,C}) = \sum_{k=1}^{C} \gamma_k \, \phi_i(\mathcal{D}_k)$ (right of Fig. 4). Then we can see that MSVs are indeed consistent with F1 score (a criterion especially suited for imbalanced data) *without* explicitly using F1 score in the computation. In addition, for KDD, due to the high class imbalance, there are classes with extremely small data size (i.e., $\leq 5$) and our calculation of $d_{\mathrm{H}}(\mathbb{Q}_i, \mathbb{Q}_{i'}; \{\mathcal{D}_k\})$ naturally suppresses their effect (possibly inaccurate $\mathbb{Q}_i$) via $\gamma_k := |\mathcal{D}_k|$. However, in practice, it should be noted that partitioning the query set to obtain a lower level of Dirichlet abstraction should be considered w.r.t. the size of available query set (Sec. 2). In other words, to obtain a lower level of Dirichlet abstraction (and thus a more refined representation), it incurs a higher cost from collecting a larger query set. In our experiments, we find that the size of each partitioned query set $\mathcal{D}_k$ should contain at least $10^2$ samples (e.g., for KDD most classes have at least or close to $10^2$ samples).

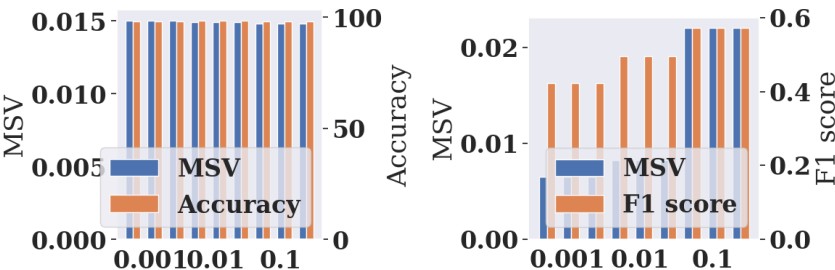

Figure 4: $\phi_i(\mathcal{D})$ (left) and $\phi_i(\{\mathcal{D}_k\})$ (right) vs. the sizes of training data (as a proportion to the full dataset).

**Identifying valuable models to purchase.** For a more end-to-end use case (instead of a single evaluation criterion), we evaluate the MSVs for up to $50$ models and the performance of a larger

learner by including a subset of these models based on their MSVs in a highest/lowest-first sequence in Fig. 5. The larger learner is random forests (voting classifier) and models are decision trees (LeNets [46]) for Breast Cancer (CIFAR-10). As the test accuracy of highest MSV-first increases more quickly (orange line), it verifies our previous comment on models that perform well when combined with other models are likely to have high MSVs. This characteristic offers some practical utility. If a buyer is looking to purchase models from a marketplace [66], then following the highest MSV sequence, the buyer only needs to purchase a subset of 15 (left of Fig. 5) or 25 (right of Fig. 5) out all 50 available models, thus saving cost. More results on CIFAR-100 with ResNet-18 are in App. C.

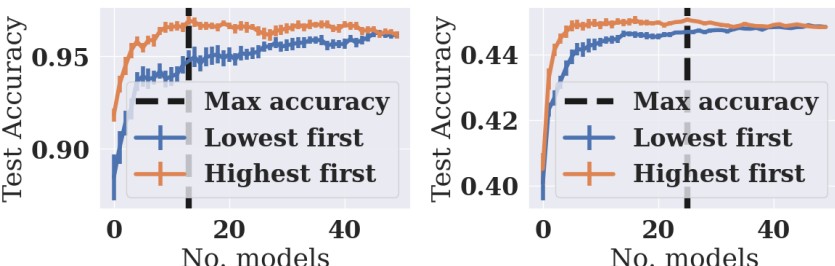

Figure 5: Test accuracy vs. number of models on Breast Cancer dataset (left) and CIFAR-10 (right).

# 6   Related Work

The work of [66] investigates the binary classification setting and is not suitable for empirical comparison as we consider problems with multiple classes. The works of [10, 11, 13, 51] approach the design of a model marketplace from an economics perspective by addressing issues like arbitrage (e.g., via horizontal/vertical pricing). In contrast, we formalize the value of a model via what it has learned w.r.t. a task. The black-box access setting is appealing in a model marketplace as it accommodates different model types. Some existing methods [8, 29, 30, 33, 44] focus on how to learn a fused model from several trained models (possibly with black-box access) instead of how to value these models. We highlight that we design the fusion (Definition 3) to leverage its analytic properties in Theorem 1. The approach of learning the Shapley value arises in data valuation problems but has not been considered in model valuation. Interestingly, we can draw parallels between Theorem 1 and [19, Theorem 2.8]. For brevity, we include a more extensive contrasting comparison with data valuation in App. B.

# 7   Discussion and Future Work

We exploit a *Dirichlet abstraction* of classification models with only black-box access for proposing a novel equitable model valuation called the *model Shapley*. We discuss that choosing a suitable level of the Dirichlet abstraction can improve how accurately MSV reflects a model's predictive performance and empirically show that using the partitioned query sets (according to the classes) can provide a suitable trade-off between the level of abstraction and the size of the available query set. MSV behaves consistently (in our experiments) with some common model evaluation criteria (i.e., predictive accuracy and certainty, F1 score) and can be extended to more sophisticated criteria. This implies MSV can potentially help unify existing evaluation criteria to provide a simplified model valuation in practice, without needing to explicitly perform separate evaluations.

For future work, it is interesting to explore how model valuation can help address the practical considerations encountered in existing data valuation methods [70, 78, 80] and to apply this technique to existing collaborative (learning) frameworks which require a valuation of models, both non-parametric ones [2, 62, 72, 74, 81] and parameterized ones [18, 49, 79]. Moreover, a more detailed investigation into satisfying the equitability of Shapley value [59] and its trade-off with the computational cost [82] is of practical interest, such as by applying more sophisticated analyses [47] or methods [9, 31, 32, 52] for our proposed Gaussian process regression learning approach.

## Acknowledgments and Disclosure of Funding

This research/project is supported by the National Research Foundation Singapore and DSO National Laboratories under the AI Singapore Programme (AISG Award No: AISG2-RP-2020-018). Xinyi Xu is supported by the Institute for Infocomm Research of Agency for Science, Technology and Research (A*STAR).

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

## A   Theoretical Discussion

### A.1   Results Related to Maximum Likelihood Estimation (MLE)

**Log-likelihood function $F$ (1).**   For completeness, we provide the full log-likelihood function [36, 57, 65] using the predictions as follows: To avoid notational overload, let $y_j := \mathbf{M}_i(x_j)$ denote the predictive probability vector of $\mathbf{M}_i$ on $x_j$ and its dependence on $i$ is suppressed. Similarly, we suppress the dependence of $\boldsymbol{\alpha}$ on $i$ since the context is clear. Then,

$$F(\boldsymbol{\alpha}, \{y_j\}_{j=1,\dots,D}) = \log p(\{y_j\}_{j=1,\dots,D}|\boldsymbol{\alpha}) = \log \prod_{j=1}^{D} p(y_j|\boldsymbol{\alpha})$$

$$= \log \prod_{j=1}^{D} \frac{\Gamma(\sum_{k=1}^{C} \alpha_k)}{\prod_{k=1}^{C} \Gamma(\alpha_k)} \prod_{k=1}^{C} y_{j,k}^{\alpha_k - 1}$$

$$= D \left[ \log \Gamma \left( \sum_{k=1}^{C} \alpha_k \right) - \sum_{k=1}^{C} \log \Gamma(\alpha_k) + \sum_{k=1}^{C} (\alpha_k - 1) \log \bar{h}_{i,k} \right]$$

where $\log \bar{h}_{i,k}$ is the $k$-th component of $\log \bar{h}_i$. Since the final expression of the above log-likelihood function $F$ only depends on the log-predictions $\{\log y_j\}_{j=1,\dots,D}$ through the observed sufficient statistics $\log \bar{h}_i$ (i.e., with an element-wise log operation), we can directly use $\bar{h}_i$ in (1) instead of $\{y_j\}_{j=1,\dots,D}$.

**Similar predictions between models imply similar MLE approximations of their Dirichlet abstractions.**   Intuitively, two models produce similar predictions if and only if their Dirichlet abstractions are similar. The result below precisely formalizes this intuition by exploiting the analytic tractability of the log-likelihood function $F$ (1) which is concave with a unique maximizer:

**Proposition 3.** Suppose that the observed sufficient statistics $\log \bar{h}_i$ ($\log \bar{h}_{i'}$) based on the predictions of $\mathbf{M}_i$ ($\mathbf{M}_{i'}$) on $\mathcal{D}$ and the MLE approximation of its Dirichlet abstraction $\boldsymbol{\alpha}_i$ ($\boldsymbol{\alpha}_{i'}$) are given. Then,

$$|\bar{h}_i - \bar{h}_{i'}|_1 = 0 \iff \boldsymbol{\alpha}_i = \boldsymbol{\alpha}_{i'} .$$

*Proof.* From (1),

$$F(\boldsymbol{\alpha}, \bar{h}_i) = D \times [G(\boldsymbol{\alpha}) + \log \bar{h}_i \, \boldsymbol{\alpha}^\top - \log \bar{h}_i \, \mathbf{1}^\top]$$

$$\underset{\boldsymbol{\alpha}}{\operatorname{argmax}} F(\boldsymbol{\alpha}, \bar{h}_i) = \underset{\boldsymbol{\alpha}}{\operatorname{argmax}} \, G(\boldsymbol{\alpha}) + \log \bar{h}_i \, \boldsymbol{\alpha}^\top$$

$$= \underset{\boldsymbol{\alpha}}{\operatorname{argmax}} \, G(\boldsymbol{\alpha}) + \log \bar{h}_{i'} \, \boldsymbol{\alpha}^\top - \log \bar{h}_{i'} \, \boldsymbol{\alpha}^\top + \log \bar{h}_i \, \boldsymbol{\alpha}^\top$$

$$= \underset{\boldsymbol{\alpha}}{\operatorname{argmax}} \, F(\boldsymbol{\alpha}, \bar{h}_{i'}) + D \left( \log \bar{h}_i - \log \bar{h}_{i'} \right) \boldsymbol{\alpha}^\top . \quad (4)$$

Note that the RHS expression of (4) to be maximized remains concave since $F(\boldsymbol{\alpha}, \bar{h}_{i'})$ is concave in $\boldsymbol{\alpha}$ [36] and the $D \left( \log \bar{h}_i - \log \bar{h}_{i'} \right) \boldsymbol{\alpha}^\top$ term is linear in $\boldsymbol{\alpha}$. If $|\bar{h}_i - \bar{h}_{i'}|_1 = 0$, then the $D \left( \log \bar{h}_i - \log \bar{h}_{i'} \right) \boldsymbol{\alpha}^\top$ term in (4) becomes 0. So, $\boldsymbol{\alpha}_i = \operatorname{argmax}_{\boldsymbol{\alpha}} F(\boldsymbol{\alpha}, \bar{h}_i) = \operatorname{argmax}_{\boldsymbol{\alpha}} F(\boldsymbol{\alpha}, \bar{h}_{i'}) = \boldsymbol{\alpha}_{i'}$ since $F$ is concave with a unique maximizer. Therefore, $|\bar{h}_i - \bar{h}_{i'}|_1 = 0 \implies \boldsymbol{\alpha}_i = \boldsymbol{\alpha}_{i'}$.

From Lemma 1 below, $\log \bar{h}_{i,k} = \psi(\alpha_{i,k}) - \psi(|\boldsymbol{\alpha}_i|_1)$ and $\log \bar{h}_{i',k} = \psi(\alpha_{i',k}) - \psi(|\boldsymbol{\alpha}_{i'}|_1)$ where $\alpha_{i,k}$ ($\alpha_{i',k}$) is the $k$-th component of $\boldsymbol{\alpha}_i$ ($\boldsymbol{\alpha}_{i'}$) and $\psi$ is the digamma function.[9] It follows immediately that

$$\bar{h}_{i,k} - \bar{h}_{i',k} = \exp[\psi(\alpha_{i,k}) - \psi(|\boldsymbol{\alpha}_i|_1)] - \exp[\psi(\alpha_{i',k}) - \psi(|\boldsymbol{\alpha}_{i'}|_1)] ,$$

so $\boldsymbol{\alpha}_i = \boldsymbol{\alpha}_{i'} \implies |\bar{h}_i - \bar{h}_{i'}|_1 = 0$. □

Importantly, both abstractions are able to preserve the similarity between $\mathbf{M}_i$ and $\mathbf{M}_{i'}$, via either the Hellinger distance $d_{\mathrm{H}}(\boldsymbol{\alpha}_i, \boldsymbol{\alpha}_{i'})$ (further discussed in Sec. 4.1) or the $\ell_1$ distance $|\bar{h}_i - \bar{h}_{i'}|_1$

---

[9]The digamma function $\psi(x) := \mathrm{d}/\mathrm{d}x(\ln \Gamma(x))$ is a monotonically increasing function that converges to $\ln x - 1/2x$ for any $x > 0$.

(Proposition 3 above) where a lower $\ell_1$ or $d_H$ distance is equivalent to a higher similarity. We also empirically compare $\boldsymbol{\alpha}_i, \bar{h}_i$ in Sec. 4.2 and find that in general $\boldsymbol{\alpha}_i$ is better.

**Lemma 1.** Suppose that the observed sufficient statistics $\log \bar{h}_i$ based on the predictions of $\mathbf{M}_i$ on $\mathcal{D}$ and the MLE approximation of its Dirichlet abstraction $\boldsymbol{\alpha}_i$ are given. Then,

$$\log \bar{h}_{i,k} = \psi(\alpha_{i,k}) - \psi(|\boldsymbol{\alpha}_i|_1) .$$

*Proof.* The partial derivative of $F(\boldsymbol{\alpha}, \bar{h}_i)$ w.r.t. $\alpha_k$ can be explicitly derived as follows:

$$\frac{\partial F(\boldsymbol{\alpha}, \bar{h}_i)}{\partial \alpha_k} = D\left(\psi\left(\sum_{k=1}^{C} \alpha_k\right) - \psi(\alpha_k) + \log \bar{h}_{i,k}\right) .$$

For a concave optimization problem (i.e., maximizing the log-likelihood) subject to a non-negative orthant constraint (i.e., $\boldsymbol{\alpha} \succeq 0$), the work of [6] has shown that the optimality conditions can be expressed as

$$\nabla F(\boldsymbol{\alpha}; \bar{h}_i) \succeq 0 \quad \text{and} \quad \alpha_k \times \frac{\partial F(\boldsymbol{\alpha}, \bar{h}_i)}{\partial \alpha_k} = 0 \quad \text{for } k = 1, \ldots, C .$$

The last condition is known as the *complementarity*. Focusing on the complementarity condition w.r.t. the $k$-th components of $\boldsymbol{\alpha}_i$ and $\bar{h}_i$ gives

$$\alpha_{i,k} = 0 \quad \vee \quad \frac{\partial F(\boldsymbol{\alpha}, \bar{h}_i)}{\partial \alpha_k} = 0 .$$

Since $\alpha_{i,k} > 0$ (Definition 1),

$$\frac{\partial F(\boldsymbol{\alpha}, \bar{h}_i)}{\partial \alpha_k} = D\left(\psi(|\boldsymbol{\alpha}_i|_1) - \psi(\alpha_{i,k}) + \log \bar{h}_{i,k}\right) = 0$$

which is simplified to

$$\psi(|\boldsymbol{\alpha}_i|_1) - \psi(\alpha_{i,k}) + \log \bar{h}_{i,k} = 0$$

and thus,

$$\log \bar{h}_{i,k} = \psi(\alpha_{i,k}) - \psi(|\boldsymbol{\alpha}_i|_1) .$$

$\square$

## A.2 Results Related to Dirichlet Distribution and Distributional Distance

**Suitable measures of distance between Dirichlet distributions.** We highlight the challenges in applying the distance measures beside the Hellinger distance in App. A. Specifically, the Hellinger distance provides unique and desirable theoretical properties (e.g., satisfying the triangle inequality) which we exploit (e.g., to derive Theorem 1)

In our context, we seek a distance measure that is well-defined between two Dirichlet distributions, can be evaluated in closed form, and has analytic properties. The work of [63] has painstakingly compared several probabilistic distance measures including the Kullback–Leibler (KL) and symmetric KL divergences [38, 42], Patrick–Fischer distance [60], generalized Matusita distance [4], Lissack–Fu distance [50], Kolmogorov [15] distance, Chernoff distance [12], and Hellinger distance [27], and made the following observations: The KL and symmetric KL divergences and the Patrick–Fischer distance can encounter cases where the distance becomes undefined. The generalized Matusita distance, Lissack–Fu distance, and Kolmogorov distance all lack an anti-derivative. So, we consider and compare the remaining two options: Chernoff and Hellinger distances, which are connected as follows.

**Chernoff vs. Hellinger distances.** As mentioned earlier, the Chernoff distance $d_H$ is the other theoretically appealing choice for Dirichlet distributions [63], so we discuss its connection with the Hellinger distance as follows. We start by recalling the definition for Chernoff distance and derive an analytic connection between them.

**Definition 4 (Chernoff distance [12]).** The Chernoff distance between two distributions $p$ and $q$ is $d_C(p, q; \lambda) := -\ln \int p^\lambda(x) q^{1-\lambda}(x) \, dx$ where $\lambda \in (0, 1)$. Note that $d_C(p, q; \lambda = 1/2)$ is symmetric in $p$ and $q$.

For two continuous distributions $p, q$, their Hellinger distance $d_{\mathrm{H}}(p,q)$ and their Chernoff distance (with $\lambda = 1/2$) $d_{\mathrm{C}}(p,q; \lambda = 1/2)$ are connected via the *Bhattacharyya coefficient* $\mathrm{BC}(p,q) := \int \sqrt{p(x)q(x)}dx \in [0,1]$ as follows,

$$1 - d_{\mathrm{H}}^2(p,q) = \exp(-d_{\mathrm{C}}(p,q; \lambda = 1/2))$$

by substituting the equalities

$$d_{\mathrm{H}}(p,q) = \sqrt{1 - \mathrm{BC}(p,q)}\,,$$

and

$$d_{\mathrm{C}}(p,q; \lambda = 1/2) = -\ln(\mathrm{BC}(p,q))\,.$$

As a result,

$$d_{\mathrm{H}}(p,q) = \sqrt{1 - \exp(-d_{\mathrm{C}}(p,q; \lambda = 1/2))}\,, \tag{5}$$

or equivalently,

$$d_{\mathrm{C}}(p,q; \lambda = 1/2) = -\ln(1 - d_{\mathrm{H}}^2(p,q))\,. \tag{6}$$

From Equ.(6), $d_{\mathrm{C}}$ is monotonic w.r.t. $d_{\mathrm{H}}$, but $d_{\mathrm{C}}$ has a logarithmic dependence which results in it not being a proper metric (i.e., does not satisfy the triangle inequality) unlike $d_{\mathrm{H}}$. We specifically leverage the triangle inequality to prove Theorem 1, which seems difficult to do for the Chernoff distance.

Interestingly, this logarithmic dependence turns out to be a practical advantage where $d_{\mathrm{H}}$ can run the risk of numerical overflow [63]. Our clustering experiment of the models (App. C) runs into the issue of numerical overflow of the Hellinger distance and we resort to the Chernoff distance. Our empirical results (App. C) also support this that using $d_{\mathrm{C}}$ can obtain better results than using $d_{\mathrm{H}}$.

**Proof of Proposition 2.** Continuing from our discussion in Sec. 3, $\nu_{\mathrm{CE}}(\mathcal{C}) \leq \nu_{\mathrm{H}}^2(\mathcal{C})$ follows from Lemma 2 below by requiring a sufficient condition on the differential entropy of $\mathbb{Q}_{\mathcal{C}}$, as stated in Proposition 2.[10] On the other hand, constructing a lower bound of $\nu_{\mathrm{CE}}(\mathcal{C})$ via $\nu_{\mathrm{H}}^2(\mathcal{C})$ is less direct, as can be seen from the upper bound on CE using $d_{\mathrm{H}}^2$ in Lemma 2. Deriving such a lower bound (Proposition 2) involves exploiting a property specific to Dirichlet distributions that the differential entropy is non-positive and bounded from above.

*Proof of Proposition 2.* By substituting $p = \mathbb{Q}_{\mathcal{C}}$ and $q = \mathbb{Q}^*$ into the first inequality of Lemma 2, $2d_{\mathrm{H}}^2(\mathbb{Q}_{\mathcal{C}}, \mathbb{Q}^*) \leq \mathrm{CE}(\mathbb{Q}_{\mathcal{C}}, \mathbb{Q}^*) - H(\mathbb{Q}_{\mathcal{C}})$. Using $H(\mathbb{Q}_{\mathcal{C}}) + d_{\mathrm{H}}^2(\mathbb{Q}_{\mathcal{C}}, \mathbb{Q}^*) \geq 0$, it follows that $d_{\mathrm{H}}^2(\mathbb{Q}_{\mathcal{C}}, \mathbb{Q}^*) \leq \mathrm{CE}(\mathbb{Q}_{\mathcal{C}}, \mathbb{Q}^*)$, so $\nu_{\mathrm{CE}}(\mathcal{C}) \leq \nu_{\mathrm{H}}^2(\mathcal{C})$ by plugging in the definitions of $\nu_{\mathrm{CE}}(\mathcal{C}) := -\mathrm{CE}(\mathbb{Q}_{\mathcal{C}}, \mathbb{Q}^*)$ and $\nu_{\mathrm{H}}^2(\mathcal{C}) := -d_{\mathrm{H}}^2(\mathbb{Q}_{\mathcal{C}}, \mathbb{Q}^*)$.

Next, the differential entropy of a Dirichlet distribution $p$ parameterized by $\boldsymbol{\alpha}$ [48, Table 2.1] is

$$H(p) = -G(\boldsymbol{\alpha}) + \left(\sum_{k=1}^{C} \alpha_k - C\right) \psi\left(\sum_{k=1}^{C} \alpha_k\right) - \sum_{k=1}^{C}(\alpha_k - 1)\, \psi(\alpha_k)$$

which is maximized at $\boldsymbol{\alpha} = \mathbf{1}_C$ [55, 54], and its maximum value is therefore $-\log\Gamma(C)$. Substituting $H(p) \leq -\log\Gamma(C)$ into the second inequality of Lemma 2 gives

$$\mathrm{CE}(p,q) \leq \mathrm{const}_q \times \left(1 - [1 - d_{\mathrm{H}}^2(p,q)]^2\right) - \log\Gamma(C)\,.$$

By substituting $p = \mathbb{Q}_{\mathcal{C}}$ and $q = \mathbb{Q}^*$ into the above and plugging in the definitions of $\nu_{\mathrm{CE}}(\mathcal{C}) := -\mathrm{CE}(\mathbb{Q}_{\mathcal{C}}, \mathbb{Q}^*)$ and $\nu_{\mathrm{H}}^2(\mathcal{C}) := -d_{\mathrm{H}}^2(\mathbb{Q}_{\mathcal{C}}, \mathbb{Q}^*)$,

$$-\nu_{\mathrm{CE}}(\mathcal{C}) \leq \mathrm{const}_{\mathbb{Q}^*} \times \left(1 - [1 + \nu_{\mathrm{H}}^2(\mathcal{C})]^2\right) - \log\Gamma(C)\,,$$

which can be rearranged to complete the proof. $\qquad\square$

**Lemma 2.** For any two continuous distributions $p$ and $q$,

$$2d_{\mathrm{H}}^2(p,q) \leq \mathrm{CE}(p,q) - H(p) \leq \mathrm{const}_q \times \left(1 - [1 - d_{\mathrm{H}}^2(p,q)]^2\right)$$

where $H(p) := -\int p(x)\log p(x)\,\mathrm{d}x$ is the differential entropy of $p$, $d_{\mathrm{H}}(p,q) \in [0,1]$ is the Hellinger distance, and $\mathrm{const}_q := [\log(1/q_{\min} - 1)]/(1 - 2q_{\min})$ with $q_{\min} := \min_z q(z)$.

---

[10]Note $\nu_{\mathrm{H}}^2(\mathcal{C}) := -d_{\mathrm{H}}^2(\mathbb{Q}_{\mathcal{C}}, \mathbb{Q}^*)$.

*Proof.* Firstly,

$$\mathrm{CE}(p,q) := -\int p(x)\log q(x)\,\mathrm{d}x$$

$$= d_{\mathrm{KL}}(p,q) + H(p) \tag{7}$$

where $d_{\mathrm{KL}}(p,q) := \int p(x)\log(p(x)/q(x))\,\mathrm{d}x$ is the Kullback-Leibler distance. Next, we derive a lower bound of $d_{\mathrm{KL}}(p,q)$ in terms of $d_{\mathrm{H}}(p,q)$:

$$d_{\mathrm{KL}}(p,q) \geq 2d_{\mathrm{H}}^2(p,q) . \tag{8}$$

To prove (8),

$$\begin{aligned}
d_{\mathrm{KL}}(p,q) &= \int p(x)\log\frac{p(x)}{q(x)}\,\mathrm{d}x \\
&= 2\int p(x)\log\frac{\sqrt{p(x)}}{\sqrt{q(x)}}\,\mathrm{d}x \\
&= 2\int p(x)\left(-\log\frac{\sqrt{q(x)}}{\sqrt{p(x)}}\right)\mathrm{d}x \\
&\geq 2\int p(x)\left(1-\frac{\sqrt{q(x)}}{\sqrt{p(x)}}\right)\mathrm{d}x \\
&= \int\left\{p(x)+p(x)-2\sqrt{p(x)}\sqrt{q(x)}\right\}\mathrm{d}x \\
&= 1+\int\left\{p(x)-2\sqrt{p(x)}\sqrt{q(x)}\right\}\mathrm{d}x \\
&= \int q(x)\,\mathrm{d}x + \int\left\{p(x)-2\sqrt{p(x)}\sqrt{q(x)}\right\}\mathrm{d}x \\
&= \int\left\{q(x)+p(x)-2\sqrt{p(x)}\sqrt{q(x)}\right\}\mathrm{d}x \\
&= \int\left\{\sqrt{p(x)}-\sqrt{q(x)}\right\}^2\mathrm{d}x \\
&= 2d_{\mathrm{H}}^2(p,q)
\end{aligned}$$

where the inequality is due to $-\log z \geq 1-z$ for all $z \geq 0$ by setting $z = \sqrt{q(x)}/\sqrt{p(x)} \geq 0$. Moreover, we have an upper bound of $d_{\mathrm{KL}}(p,q)$ from [61, Equation 7.27]:

$$d_{\mathrm{KL}}(p,q) \leq \mathrm{const}_q \times \left(1-[1-d_{\mathrm{H}(p,q)}^2]^2\right) . \tag{9}$$

Lastly, substituting $d_{\mathrm{KL}}(p,q) = \mathrm{CE}(p,q) - H(p)$ from (7) into (8) and (9) completes the proof.

$\square$

**An example on how cross-entropy loss encodes the predictive accuracy and certainty.** The CE loss is used to construct upper and lower bounds for our proposed method (i.e., $\nu$ in Eq. (3)). Hence, our proposed method also encodes the predictive accuracy and certainty, as exemplified below.

Recall that the CE loss of a $C$-dimensional predicted probability vector $\hat{y}$ w.r.t. the one-hot encoded true label $y$:

$$-\sum_{k=1}^{C} y_k \times \ln(\hat{y}_k) .$$

W.l.o.g., assume that $y_1 = 1$ (i.e., the correct class is the first class).

1. For two predictions $[0.9, 0.1, 0, \ldots, 0]$ vs.$[0.1, 0.9, 0, \ldots, 0]$. The CE losses are $0.105$ and $2.30$, respectively. Note that the first prediction is correct while the second in incorrect and that both predictions are "equally certain". Hence, *a higher predictive accuracy implies a lower CE.*

2. For two predictions $[0.9, 0.1, 0, \ldots, 0]$ vs.$[0.6, 0.4, 0, \ldots, 0]$. The CE losses are $0.105$ and $0.511$, respectively. Note that both predictions are correct while the first prediction is "more certain". Hence, *a higher predictive certainty implies a lower CE*, if the prediction is correct.

**Extension to other model evaluation criteria.** In addition to predictive accuracy and certainty, there are other possible desirable criteria for model valuation such as adversarial robustness [43], distributional robustness [67], and algorithmic fairness (i.e., by removing prediction bias) in ML [56]. Put differently, it is possible to devise other model valuations based on adversarial robustness, distributional robustness, or the algorithmic fairness of a model. Interestingly, since these more sophisticated criteria all utilize the same building blocks (i.e., the predictive accuracy and certainty of models on *carefully selected* query sets), our proposed approach can subsume these different criteria, as described below and summarized in Table 1. To be a little technical, the high-level idea is that, since these criteria leverage the CE loss $\ell_{\text{CE}}$ in very specific ways, and we have derived the relationship between $d_{\text{H}}$ and CE above (Lemma 2), our approach can be extended to incorporate these criteria through careful choices of $\nu$.

For instance, the work of [43] explicitly defines "adversarial" training examples $\mathcal{D}_{\text{adv}}$ to be distinguished from "clean" training examples $\mathcal{D}_{\text{clean}}$ and evaluates a model's performance (i.e., a linear combination of the CE losses w.r.t. the adversarial and clean training examples separately) as follows:

$$\text{objective}_{\text{adv}} := \ell_{\text{CE}}(\mathbf{M}; \mathcal{D}_{\text{adv}}) + \upsilon \, \ell_{\text{CE}}(\mathbf{M}; \mathcal{D}_{\text{clean}})$$

for some weight $\upsilon > 0$ where we suppress the constants that linearly depend on the sizes of $\mathcal{D}_{\text{adv}}$ and $\mathcal{D}_{\text{clean}}$ for simplicity and $\ell_{\text{CE}}(\mathbf{M}; \mathcal{D})$ is the common CE loss incurred by the predictions of $\mathbf{M}$ on query set $\mathcal{D}$.

The work of [67, Equation 5] gives the *group-adjusted* distributionally robust optimization (DRO) estimator where each "group" $g \in \mathcal{G}$ contains training examples from a possibly different data distribution. Effectively, the optimizer minimizes the maximum CE loss over different groups/query sets s.t. each query set is a dataset from a possibly different data distribution:

$$\text{objective}_{\text{DRO}} := \max_{g \in \mathcal{G}} \ell_{\text{CE}}(\mathbf{M}; \mathcal{D}_{\text{g}})$$

where, for simplicity, a group size-dependent constant and a model capacity-dependent constant are ignored.

The work of [56] presents a number of different definitions of fairness in ML to cater to different situations. In general, these definitions each describe a particular way for the model to make classifications w.r.t. specific conditions on (the features of) the data in order to be fair. For simplicity, we illustrate with *equal opportunity* (EO) [56, Definition 2] which "means that the probability of a person in a positive class being assigned to a positive outcome should be equal for both protected and unprotected (female and male) group members." To relate this to CE, we can define two query sets: $\mathcal{D}_{\text{prot}}^+$ containing positive training examples under the protected group and $\mathcal{D}_{\text{unprot}}^+$ containing positive training examples under the unprotected group. To achieve equal opportunity, the average CE losses on both query sets should be (approximately) equal (i.e., both groups have equal true positive rates) or, equivalently, the difference in the CE losses should be small:

$$\text{objective}_{\text{EO}} := \left| \ell_{\text{CE}}(\mathbf{M}; \mathcal{D}_{\text{prot}}^+) - \ell_{\text{CE}}(\mathbf{M}; \mathcal{D}_{\text{unprot}}^+) \right| .$$

Table 1 gives the specific definitions of query sets with the corresponding (possible) choices of the characteristic function $\nu$ adapted from the above-mentioned minimization objectives by replacing $\ell_{\text{CE}}(\mathbf{M}; \mathcal{D})$ with $d(\mathbb{Q}, \mathbb{Q}^*; \mathcal{D})$ and adding a negation since these are minimization objectives.

Firstly, we show the original implementations/formulations using the CE loss $\ell_{\text{CE}}$ [43, 67, 56] can be reformulated using the CE between Dirichlet abstractions. We provide the reformulation of $\text{objective}_{\text{DRO}}$,

$$\max_{g \in \mathcal{G}} \text{CE}(\mathbb{Q}, \mathbb{Q}^*; \mathcal{D}) ,$$

by replacing $\ell_{\text{CE}}$ with CE on the respective Dirichlet abstraction $\mathbb{Q}$ (for $\mathbf{M}$) and an expert $\mathbb{Q}^*$ (from the test set) and omit the explicit derivations of the other two for brevity.

Such reformulation is enabled by the connection between CE and $\ell_{\text{CE}}$: The CE loss $\ell_{\text{CE}}(\mathbf{M}; \mathcal{D})$ implicitly assumes an expert who provides the correct labels to compute the loss on the query set $\mathcal{D}$.

On the other hand, $\mathrm{CE}(\mathbb{Q}, \mathbb{Q}^*; \mathcal{D})$ explicitly uses the expert and measures the cross-entropy between the two Dirichlet abstractions (i.e., one for the model $\mathbf{M}$ and the other for the expert) on the query set $\mathcal{D}$. Therefore, if a model $\mathbf{M}$ makes predictions similar to the expert on a fixed query set $\mathcal{D}$, then both $\ell_{\mathrm{CE}}(\mathbf{M}; \mathcal{D})$ and $\mathrm{CE}(\mathbb{Q}, \mathbb{Q}^*; \mathcal{D})$ are small (i.e., optimum is 0). Next, inspired by the relationship between $d_{\mathrm{H}}$ and CE in Proposition 2, the expressions in CE are reformulated using $d_{\mathrm{H}}$ in Table 1. Note that Proposition 2 provides us with the intuition and is not used exactly. Hence, our approach of decoupling the query set(s) from the model valuation makes it general enough to subsume these more sophisticated criteria (of a model's performance) for model valuation. A question one might ask is that: (How) can multiple such evaluation criteria be combined? The answer is yes, by leveraging (P3) to *linearly combine* selected evaluation criteria, as discussed next.

**Combining multiple evaluation criteria.** Specifically, for a user (e.g., potential buyer of the model) who knows the relative importance of several different criteria (formalized by the specific query sets such as in Table 1), then the user can specify the weights to achieve a desirable trade-off. This is because different users might have different preferences and there is no one-size-fits-all solution. To elaborate, suppose the user only cares about whether the model makes accurate predictions but not at all about adversarial robustness because the user intends to deploy it in a controlled and safe environment, then the task constructed for adversarial robustness is not very relevant to this user. In contrast, if the user does care about the adversarial robustness (which, is often at trade-off against pure predictive performance), then the user can set the weights between the two tasks according to their preferences.

On the other hand, if the trade-offs of the tasks are unknown, for instance the objectives are very complex, then uncovering the relationship between tasks (which are potentially trade-offs of each other) is useful. Specifically, the approach to obtain the connections in Table 1 is useful. For instance, predictive accuracy and adversarial robustness are trade-offs of each other since the objective of adversarial robustness "balances" between the clean and adversarial cross entropy (CE) losses. Upon identifying this theoretical connection, the user can then specify the weight between the two accordingly.

**Remark 1.** Note that our discussion on how to extend and combine multiple model evaluation criteria aims to provide the technical tools for doing so, instead of identifying how a buyer or a seller should use them. To elaborate, the buyer can use our method to combine several evaluation criteria based on known preferences of the relative importance of these criteria. Our discussion does not aim to guide the buyer in identifying such preferences or consider the potential asymmetry of information in a marketplace where only the buyer (or the seller) knows such preferences, how the other party should react. We believe these are further and interesting questions for future exploration.

**Other useful technical results.**

**Lemma 3 (Precision-weighted fusion preserves Dirichlet).** The precision-weighted fusion in Definition 3 follows a Dirichlet distribution: $\mathbb{Q}_{\mathcal{C}} = \mathrm{Dir}([\sum_{i=1}^n \alpha_{i,1}, \ldots, \sum_{i=1}^n \alpha_{i,C}])$ [34, Theorem 2.1].

**Lemma 4 (Bhattacharyya coefficient between Dirichlet distributions** [63]**).** Let $\boldsymbol{\alpha}$ and $\boldsymbol{\alpha}'$ denote the $C$-dimensional parameters specifying the two Dirichlet distributions $p$ and $q$, respectively. Then, the Bhattacharyya coefficient is

$$\mathrm{BC}(p,q) = \frac{\prod_{k=1}^C \Gamma((\alpha_k + \alpha_k')/2)}{\Gamma(\sum_{k=1}^C (\alpha_k + \alpha_k')/2)} \times \frac{\sqrt{\Gamma(|\boldsymbol{\alpha}|_1)\,\Gamma(|\boldsymbol{\alpha}'|_1)}}{\sqrt{\prod_{k=1}^C \Gamma(\alpha_k)\,\Gamma(\alpha_k')}}\,.$$

**Lemma 5.** The Hellinger distance between two Dirichlet distributions $p$ and $q$ parameterized by the respective $\boldsymbol{\alpha}$ and $\boldsymbol{\alpha}'$ is

$$d_{\mathrm{H}}(p,q) = \sqrt{2} \times \left(1 - \frac{\prod_{k=1}^C \Gamma((\alpha_k + \alpha_k')/2)}{\Gamma(\sum_{k=1}^C (\alpha_k + \alpha_k')/2)} \times \frac{\sqrt{\Gamma(|\boldsymbol{\alpha}|_1)\,\Gamma(|\boldsymbol{\alpha}'|_1)}}{\sqrt{\prod_{k=1}^C \Gamma(\alpha_k)\,\Gamma(\alpha_k')}}\right)^{1/2},$$

which follows directly from an equivalent definition of $d_{\mathrm{H}}(p,q) := \sqrt{2} \times \sqrt{1 - \mathrm{BC}(p,q)}$ .

### A.3 Lipschitz Continuity of Model Shapley

We provide the proof of Theorem 1 here and describe a sufficient condition for fusion (i.e., Definition 3) to increase similarity.

**Further elaboration on Lipschitz continuity.** We adopt the following general definition for Lipschitz continuity: For two metric spaces $(\mathbb{X}, d_\mathbb{X})$ and $(\mathbb{Y}, d_\mathbb{Y})$, a function $f : \mathbb{X} \mapsto \mathbb{Y}$ is $L_f$-Lipschitz continuous if there exists a constant $L_f > 0$ s.t. $\forall x_1, x_2 \in \mathbb{X}$

$$d_\mathbb{Y}(f(x_1), f(x_2)) \leq L_f d_\mathbb{X}(x_1, x_2) \ .$$

In our formulation, for the model Shapley function $\Phi$, the input space is the set $[N]$ (or more precisely the set $\{\mathbb{Q}_i : i \in [N]\}$ of Dirichlet abstractions) of the $N$ models and the metric is the Hellinger distance $d_\mathrm{H}$, which is a proper metric for probability distributions (in this case Dirichlet distributions); the output space is $\mathbb{R}$ (i.e., for model Shapley values) and the metric is the absolute difference (i.e., $|\phi_i - \phi_{i'}|$ for two inputs $\mathbb{Q}_i, \mathbb{Q}_{i'}$).

In summary, recall that $\phi_i \leftarrow \Phi(i, \nu, \{\mathbf{M}_i\}_{i \in [N]})$ in Sec. 3, the Lipschitz continuity of $\Phi$ is w.r.t. its first argument (i.e., $i$), when $\nu$ is defined as in (3) and the set $\{\mathbf{M}_i\}_{i \in [N]}$ of models is fixed (i.e., correspondingly the set $\{\mathbb{Q}_i\}_{i \in [N]}$ is also fixed). For brevity, we suppress the notational dependence on the latter arguments and write as $\phi_i := \Phi(i; \cdot) : [N] \mapsto \mathbb{R}$ where as mentioned above, the metric for the input space is defined as $d(i, i') := d_\mathrm{H}(\mathbb{Q}_i, \mathbb{Q}_{i'})$ for $i, i' \in [N]$ and the metric for the outputs is the absolute difference $|\phi_i - \phi_{i'}|$ .

**Proof of Theorem 1.** We follow an idea that the similarity between the Dirichlet abstractions $\mathbb{Q}_i$ and $\mathbb{Q}_{i'}$ will lead to a small difference in their expected marginal contributions $\phi_i$ and $\phi_{i'}$ when fused with a common $\mathbb{Q}_\mathcal{C}$ for any $\mathcal{C} \subseteq [N] \setminus \{i, i'\}$. Consequently, we can apply Lemma 6 in App. A.4.

Firstly, from $\nu_\mathrm{H}(\mathcal{C}) = -d_\mathrm{H}(\mathbb{Q}_\mathcal{C}, \mathbb{Q}^*)$ as in (3),

$$\nu_\mathrm{H}(\mathcal{C} \cup \{i\}) - \nu_\mathrm{H}(\mathcal{C} \cup \{i'\}) = -d_\mathrm{H}(\mathbb{Q}_{\mathcal{C} \cup \{i\}}, \mathbb{Q}^*) + d_\mathrm{H}(\mathbb{Q}_{\mathcal{C} \cup \{i'\}}, \mathbb{Q}^*) \ .$$

Then, using the property of triangle inequality of $d_\mathrm{H}$, it follows that

$$| - d_\mathrm{H}(\mathbb{Q}_{\mathcal{C} \cup \{i\}}, \mathbb{Q}^*) + d_\mathrm{H}(\mathbb{Q}_{\mathcal{C} \cup \{i'\}}, \mathbb{Q}^*)| \leq d_\mathrm{H}(\mathbb{Q}_{\mathcal{C} \cup \{i\}}, \mathbb{Q}_{\mathcal{C} \cup \{i'\}}) \leq d_\mathrm{H}(\mathbb{Q}_i, \mathbb{Q}_{i'})$$

where the last inequality is due to the fusion-increases-similarity condition stated in Theorem 1 (and examined below by Proposition 4). The final result can be obtained applying Lemma 6: Note that $d(i, i') := d_\mathrm{H}(\mathbb{Q}_i, \mathbb{Q}_{i'}) < 1$ (since the Hellinger distance is upper bounded by 1), so the condition in Lemma 6 is satisfied with $L = 1$. In other words, for a different distance (other than the Hellinger distance), a different value of $L$ may be necessary. The constant $Z$ is directly inherited to be the Lipschitz constant. $\qquad \square$

**Proposition 4 (Fusion increases similarity).** Suppose that $\boldsymbol{\alpha}_i$ and $\boldsymbol{\alpha}_{i'}$ parameterize $\mathbb{Q}_i$ and $\mathbb{Q}_{i'}$, respectively. Then,

$$\left[ \forall k \in [C] \quad \psi\left(\frac{\alpha_{i,k} + \alpha_{i',k}}{2}\right) - \psi(\alpha_{i,k}) \quad \geq \quad \psi\left(\frac{|\boldsymbol{\alpha}_i|_1 + |\boldsymbol{\alpha}_{i'}|_1}{2}\right) - \psi(|\boldsymbol{\alpha}_i|_1) \right.$$
$$\left. \wedge \quad \psi\left(\frac{\alpha_{i,k} + \alpha_{i',k}}{2}\right) - \psi(\alpha_{i',k}) \quad \geq \quad \psi\left(\frac{|\boldsymbol{\alpha}_i|_1 + |\boldsymbol{\alpha}_{i'}|_1}{2}\right) - \psi(|\boldsymbol{\alpha}_{i'}|_1) \right]$$
$$\implies \quad d_\mathrm{H}(\mathbb{Q}_{\mathcal{C} \cup \{i\}}, \mathbb{Q}_{\mathcal{C} \cup \{i'\}}) \leq d_\mathrm{H}(\mathbb{Q}_i, \mathbb{Q}_{i'})$$

where $\psi$ is the digamma function.

*Proof of Proposition 4.* It can be observed from Lemma 5 that $d_\mathrm{H}(p, q)$ increases iff $\mathrm{BC}(p, q)$ (Lemma 4) decreases. Then, it is equivalent to show that

$$\mathrm{BC}(\mathbb{Q}_{\mathcal{C} \cup \{i\}}, \mathbb{Q}_{\mathcal{C} \cup \{i'\}}) \geq \mathrm{BC}(\mathbb{Q}_i, \mathbb{Q}_{i'}) \ .$$

It can also be observed that BC can be viewed as a differentiable function taking in $2C$ parameters (i.e., $\boldsymbol{\alpha}_i$ and $\boldsymbol{\alpha}_{i'}$). Consider w.l.o.g. its partial derivative w.r.t. $\alpha_{i,k}$ and w.r.t. $\alpha_{i',k}$ for $k = 1, \ldots, C$:

$$\frac{\partial \mathrm{BC}}{\partial \alpha_{i,k}} = \mathrm{coeff} \times \left[ -\psi(\alpha_{i,k}) + \psi\left(\frac{\alpha_{i,k} + \alpha_{i',k}}{2}\right) - \psi\left(\frac{|\boldsymbol{\alpha}_i|_1 + |\boldsymbol{\alpha}_{i'}|_1}{2}\right) + \psi(|\boldsymbol{\alpha}_i|_1) \right]$$

$$\frac{\partial \mathrm{BC}}{\partial \alpha_{i',k}} = \mathrm{coeff} \times \left[ -\psi(\alpha_{i',k}) + \psi\left(\frac{\alpha_{i,k} + \alpha_{i',k}}{2}\right) - \psi\left(\frac{|\boldsymbol{\alpha}_i|_1 + |\boldsymbol{\alpha}_{i'}|_1}{2}\right) + \psi(|\boldsymbol{\alpha}_{i'}|_1) \right]$$

where

$$\text{coeff} = \frac{\sqrt{\Gamma(|\boldsymbol{\alpha}_i|_1)\,\Gamma(|\boldsymbol{\alpha}_{i'}|_1)}\,\sum_{k=1}^{C}\Gamma((\alpha_{i,k}+\alpha_{i',k})/2)}{2\sqrt{\prod_{k=1}^{C}\Gamma(\alpha_{i,k})\,\Gamma(\alpha_{i',k})}\,\Gamma(\sum_{k=1}^{C}(\alpha_{i,k}+\alpha_{i',k})/2)} > 0$$

due to the positivity of $\Gamma$ over the positive domain.[11]

Now, Lemma 3 implies that the fusion always increases $\alpha_{i,k}$: Since $\alpha_{i,k}$ denotes the $k$-th component of $\boldsymbol{\alpha}_i$ and $\alpha_{\mathcal{C}\cup\{i\},k}$ denotes the $k$-th component of $\boldsymbol{\alpha}_{\mathcal{C}\cup\{i\}}$, $\alpha_{\mathcal{C}\cup\{i\},k} \geq \alpha_{i,k}$. So, if $\partial\mathrm{BC}/\partial\alpha_{i,k} \geq 0$ and $\partial\mathrm{BC}/\partial\alpha_{i',k} \geq 0$ for $k = 1,\ldots,C$, then the resulting BC increases (or, equivalently, $d_{\mathrm{H}}$ decreases) after fusion.

Since coeff $> 0$,

$$-\psi(\alpha_{i,k}) + \psi\left(\frac{\alpha_{i,k}+\alpha_{i',k}}{2}\right) - \psi\left(\frac{|\boldsymbol{\alpha}_i|_1 + |\boldsymbol{\alpha}_{i'}|_1}{2}\right) + \psi(|\boldsymbol{\alpha}_i|_1) \geq 0 \implies \frac{\partial\mathrm{BC}}{\partial\alpha_{i,k}} \geq 0$$

$$-\psi(\alpha_{i',k}) + \psi\left(\frac{\alpha_{i,k}+\alpha_{i',k}}{2}\right) - \psi\left(\frac{|\boldsymbol{\alpha}_i|_1 + |\boldsymbol{\alpha}_{i'}|_1}{2}\right) + \psi(|\boldsymbol{\alpha}_{i'}|_1) \geq 0 \implies \frac{\partial\mathrm{BC}}{\partial\alpha_{i',k}} \geq 0$$

for $k = 1,\ldots,C$. So, the final result follows. $\qquad\square$

Let $D_{\mathrm{sum}} \coloneqq \sum_{k=1}^{C} \psi((\alpha_{i,k}+\alpha_{i',k})/2) - (\psi(\alpha_{i,k}) + \psi(\alpha_{i',k}))/2$ and $D_{|\cdot|} \coloneqq C[\psi((|\boldsymbol{\alpha}_i|_1 + |\boldsymbol{\alpha}_{i'}|_1)/2) - (\psi(|\boldsymbol{\alpha}_i|_1) + \psi(|\boldsymbol{\alpha}_{i'}|_1))/2]$. The sufficient condition for Proposition 4 implies that $D_{\mathrm{sum}} \geq D_{|\cdot|}$. Intuitively, $D_{\mathrm{sum}}$ is a sum of dimension-/class-wise differences between $\mathbb{Q}_i$ and $\mathbb{Q}_{i'}$ and it is large if every pair of $\alpha_{i,k}$ and $\alpha_{i',k}$ are different and relatively small (i.e., low concentration for dimension/class $k$). $D_{\mathrm{sum}}$ is likely large if the dimension $C$ is large as there are more pairs of $\alpha_{i,k}$ and $\alpha_{i',k}$ whose difference contributes towards $D_{\mathrm{sum}}$. On the other hand, $D_{|\cdot|}$ is a measure of the difference in the overall precisions of $\mathbb{Q}_i$ and $\mathbb{Q}_{i'}$. $D_{|\cdot|}$ is large if $|\boldsymbol{\alpha}_i|_1$ and $|\boldsymbol{\alpha}_{i'}|_1$ are different and have small values. $D_{|\cdot|}$ is likely small if $C$ is large. As $C$ increases, the precisions $|\boldsymbol{\alpha}_i|_1$ and $|\boldsymbol{\alpha}_{i'}|_1$ will increase, which will cause $\psi((|\boldsymbol{\alpha}_i|_1 + |\boldsymbol{\alpha}_{i'}|_1)/2)$ and $(\psi(|\boldsymbol{\alpha}_i|_1) + \psi(|\boldsymbol{\alpha}_{i'}|_1))/2$ to be very close due to the converging behavior of $\psi$.

The condition $D_{\mathrm{sum}} \geq D_{|\cdot|}$ says that if the class-wise difference between $\mathbb{Q}_i$ and $\mathbb{Q}_{i'}$ outweighs the difference in their precisions, then fusion increases similarity. Intuitively, if the 'shapes' of $\mathbb{Q}_i$ and $\mathbb{Q}_{i'}$ are very different, then fusing each with a common distribution $\mathbb{Q}_{\mathcal{C}}$ 'evens out' the difference in their shapes and increases the similarity. In particular, if $C$ is large (i.e., a high-dimensional classification task), then the condition is more likely to be satisfied.

**Empirical verification of Theorem 1.** Specifically, we verify whether a small $d_{\mathrm{H}}(\mathbb{Q}_i, \mathbb{Q}_{i'})$ leads to a small $|\phi_i - \phi_{i'}|$ via the Pearson coefficient between $d_{\mathrm{H}}(\mathbb{Q}_i, \mathbb{Q}_{i'})$ and $|\phi_i - \phi_{i'}|$ over all $i, i'$, and visualizing the corresponding heatmaps (Fig. 6). The setting is as follows. We investigate 5 real-world datasets (and various ML models), including MNIST, CIFAR-10 [41], two medical datasets: a drug reviews dataset that classifies the type of prescribed medicine based on the text reviews (DrugRe) [23] and a medical imaging dataset that classifies the medical department from the diagnostic scans (MedNIST) [58], and a cyber-threat detection dataset that classifies network intrusion based on input features such as IP addresses and network communication protocol (KDD99) [28]. Recall these are some of the highlighted application domains of model valuation (i.e., medicine and cyber-defense) in Sec. 1. Query set $\mathcal{D}$ is the respective test set of each dataset *without* partitioning.

We adopt a grouping paradigm where the grouped models have the same model type (but undergo independent training with the same data) to ensure some similarity within each group. So, we can verify whether the models within the same group have similar MSVs. To see why models of the same type can produce similar Dirichlet abstractions, we provide a clustering result of $\mathbb{Q}_i$ (Fig. 11 in App. C). Specifically, for MNIST, we implement 3 model types: logistic regression (LR), multilayer perceptron (MLP), and a 2-layer convolutional neural network (CNN). For CIFAR-10, we utilize 3 known model types with pre-trained weights: ResNet-18 [26], SqueezeNet [37], and DenseNet-121 [35]. For DrugRe, we use a CNN and a *bi-directional long-short term memory* (BiLSTM) network. More details on other datasets are in App. C.

---

[11] The code for verifying this partial derivative using an automatic differentiation package is included in the supplementary material.

The high Pearson coefficients in Table 3 provide some verification for Theorem 1. The matched color intensities in the heatmaps in Fig. 6 confirm that similar models have similar MSVs. Left (right) of Fig. 6 is w.r.t. general (class-specific) Dirichlet abstractions.

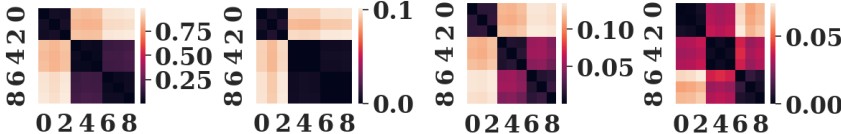

Figure 6: Left two plots are heatmaps for $d_H(\mathbb{Q}_i, \mathbb{Q}_{i'})$ and $|\phi_i - \phi_{i'}|$. Right two plots are heatmaps for $d_H(\mathbb{Q}_i, \mathbb{Q}_{i'}; \{\mathcal{D}_k\})$ and $|\phi_i(\{\mathcal{D}_k\}) - \phi_{i'}(\{\mathcal{D}_k\})|$.

Table 3: Pearson coefficient between $d_H(\mathbb{Q}_i, \mathbb{Q}_{i'})$ and $|\phi_i - \phi_{i'}|$ using the test set as a single query set. A high coefficient verifies Theorem 1.

| MNIST | CIFAR-10 | MedNIST | KDD | DrugRe |
|---|---|---|---|---|
| 0.974 | 0.996 | 0.974 | 0.999 | 0.955 |

### A.4 Model Shapley Value

**Closed-form expression of $\nu$ and the resulting computational complexity.** Recall that an advantage of the combined choices of Definition 3 and the Hellinger distance $d_H$ (described in Sec. 3) is an available closed form evaluation of $d_H$ between two Dirichlet abstractions, which has a constant time computational cost (i.e., $\mathcal{O}(1)$). This is important as $d_H$ is used to define the characteristic function $\nu$ (3) for model Shapley (2). Specifically, note that the definition of model Shapley (2) requires evaluating the characteristic function $\nu$ for an exponential number of times due to the summation over all possible subsets $\mathcal{C} \subseteq [N] \setminus \{i\}$. In other words, even with a characteristic function $\nu$ that can be evaluated in constant time (e.g., $d_H$), the computational complexity of MSV is still at least exponential in $N$, which is almost intractable for large $N$; if evaluating the characteristic function has a higher computational complexity, then the computational complexity of MSV would be even more intractable.

**Similarity-bounded difference in SVs.** The following lemma provides a general result for bounding the difference in two Shapley values (not necessarily MSVs) and is used in the proof of Theorem 1. It may be of independent interest.

**Lemma 6 (Similarity-bounded difference in the Shapley values).** For all $i, i' \in [N]$,
$$(\forall \mathcal{C} \subseteq [N] \setminus \{i, i'\} \ |\nu(\mathcal{C} \cup \{i\}) - \nu(\mathcal{C} \cup \{i'\})| \le Ld(i, i')) \implies |\phi_i - \phi_{i'}| \le ZLd(i, i')$$
where $L \ge 0$ is a constant and $d(i, i')$ is some distance measure between $i$ and $i'$ and $Z$ is the linear scaling as in (2).

*Proof of Lemma 6.* The difference $|\phi_i - \phi_i'|$ can be bounded by enumerating the coalitions in a paired way as follows:

$$|\phi_i - \phi_{i'}| \le \sum_{\mathcal{C} \subseteq [N] \setminus \{i, i'\}} Z\omega_{\mathcal{C}} |\nu(\mathcal{C} \cup \{i\}) - \nu(\mathcal{C} \cup \{i'\})| + \sum_{\substack{(\mathcal{C}, \mathcal{C}'), \\ \mathcal{C} \subseteq [N] \setminus \{i\}; i' \in \mathcal{C}, \\ \mathcal{C}' \subseteq [N] \setminus \{i'\}; i \in \mathcal{C}', \\ \mathcal{C} \cup \{i\} = \mathcal{C}' \cup \{i'\}}} Z\omega_{\mathcal{C}} |\nu(\mathcal{C}') - \nu(\mathcal{C})|$$

where $\omega_{\mathcal{C}} := |\mathcal{C}|!(N - |\mathcal{C}| - 1)!/N!$ and we have multiplied the constant $Z$ into the separate summations. Note that this enumeration (considering both summations) exhausts (w.l.o.g. from the viewpoint of $i$) $\mathcal{C} \subseteq [N] \setminus \{i\}$ in the calculation of $\phi_i$. The first summation enumerates all $\mathcal{C} \subseteq [N] \setminus \{i, i'\}$, so the remaining $\mathcal{C}$ to consider for $i$'s marginal contributions as in (2) are the ones that include $i'$ but not $i$ (considered in the second summation). The summands in the second summation are in fact also in the form of $|\nu(\mathcal{C} \cup \{i\}) - \nu(\mathcal{C} \cup \{i'\})|$ for some $\mathcal{C} \subseteq [N] \setminus \{i, i'\}$. Note that in the second summation,

$$(\mathcal{C} \cup \{i\} = \mathcal{C}' \cup \{i'\}) \wedge i' \in \mathcal{C} \wedge i \in \mathcal{C}' \ ,$$

so $\mathcal{C}' \setminus \{i\} = \mathcal{C} \setminus \{i'\}$. Consequently, let $\underline{\mathcal{C}} := \mathcal{C}' \setminus \{i\}$. Then,

$$|\nu(\mathcal{C}') - \nu(\mathcal{C})| = |\nu(\underline{\mathcal{C}} \cup \{i\}) - \nu(\underline{\mathcal{C}} \cup \{i'\})| .$$

To complete the final step, first consider the simpler case of $Z = 1$ and the coefficient $\omega_{\mathcal{C}}$ is defined in a way such that $\phi_i$ satisfies efficiency [69] (i.e., $\sum_{\mathcal{C} \subseteq [N] \setminus \{i\}} \omega_{\mathcal{C}} = 1$), the condition $\forall \mathcal{C} \subseteq [N] \setminus \{i, i'\}$ $|\nu(\mathcal{C} \cup \{i\}) - \nu(\mathcal{C} \cup \{i'\})| \leq Ld(i, i')$ can be used to bound the overall sum of the RHS as

$$|\phi_i - \phi_{i'}| \leq Ld(i, i') .$$

More generally for $Z \neq 1$, it can be directly multiplied to the RHS as follows

$$|\phi_i - \phi_{i'}| \leq ZLd(i, i')$$

since every term is multiplied by the same constant in the above two summations. $\qquad \square$

**Proposition 5** (**Diminishing Model Shapley Value due to Substitutes**). According to the definitions as in (P4), $\phi_i$ ($\phi_i'$) denotes the model Shapley value of $\mathbf{M}_i$ w.r.t. $[N]$ ($[N'] := [N] \cup \{i_{\mathrm{c}}\}$) and $\mathbf{M}_{i_{\mathrm{c}}} = \mathbf{M}_i$ is a perfect substitute/identical duplicate/copy. Then,

$$(\forall \mathcal{C} \subseteq [N] \setminus \{i\} \ \nu(\mathcal{C} \cup \{i\}) - \nu(\mathcal{C}) \geq \nu(\mathcal{C} \cup \{i, i_{\mathrm{c}}\}) - \nu(\mathcal{C} \cup \{i\})) \implies \phi_i' \leq \phi_i .$$

*Proof of Proposition 5.* The inequality $\phi_i \geq \phi_i'$ is shown by examining the pairwise difference over their respective summands in the summation of model Shapley.

For $\phi_i$, $\phi_i := \sum_{\mathcal{C} \subseteq [N] \setminus \{i\}} s_{\mathcal{C}}$ where the summand $s_{\mathcal{C}} := \omega_{\mathcal{C}} \mathrm{MC}_i(\mathcal{C})$. $\omega_{\mathcal{C}}$ is as in (2) and $\mathrm{MC}_i(\mathcal{C}) := \nu(\mathcal{C} \cup \{i\}) - \nu(\mathcal{C})$ is the marginal contribution of $i$ w.r.t. $\mathcal{C}$.

For $\phi_i'$, $\phi_i' := \sum_{\substack{\mathcal{C} \subseteq [N] \setminus \{i\} \\ \mathcal{C}' = \mathcal{C} \cup \{i_{\mathrm{c}}\}}} s_{\mathcal{C}}'$ where the summand

$$s_{\mathcal{C}}' := \frac{|\mathcal{C}|!(N + 1 - |\mathcal{C}| - 1)!}{(N + 1)!} \mathrm{MC}_i(\mathcal{C}) + \frac{|\mathcal{C}'|!(N + 1 - |\mathcal{C}'| - 1)!}{(N + 1)!} \mathrm{MC}_i(\mathcal{C}') .$$

$\phi_i'$ is obtained by observing that adding $i_{\mathrm{c}}$ to $[N]$ means additionally enumerating all the $\mathcal{C} \subseteq [N] \setminus \{i\}$ but added with $i_{\mathrm{c}}$, as shown above.

As $|\mathcal{C}'| = |\mathcal{C}| + 1$,

$$\begin{aligned}
s_{\mathcal{C}}' &= \frac{\omega_{\mathcal{C}}}{N + 1} [(N - |\mathcal{C}|) \, \mathrm{MC}_i(\mathcal{C}) + (|\mathcal{C}| + 1) \, \mathrm{MC}_i(\mathcal{C}')] \\
&\leq \frac{\omega_{\mathcal{C}}}{N + 1} [(N - |\mathcal{C}|) \, \mathrm{MC}_i(\mathcal{C}) + (|\mathcal{C}| + 1) \, \mathrm{MC}_i(\mathcal{C})] \\
&= \frac{\omega_{\mathcal{C}}}{N + 1} [(N + 1) \, \mathrm{MC}_i(\mathcal{C})] \\
&= s_{\mathcal{C}}
\end{aligned}$$

where the inequality is due to the conditionally redundant property of $\nu$.[12] Since $\phi_i$ and $\phi_i'$ enumerate the same summation and individual summands have $s_{\mathcal{C}}' \leq s_{\mathcal{C}}$, it follows that $\phi_i' \leq \phi_i$. $\qquad \square$

**Remedy for duplication from a dishonest seller.** While the marginal utility of each duplicate model decreases, the combined utility of all the duplicated models may be higher than if there is only one such model. Hence, a dishonest seller might exploit this by duplicating a model to receive a higher combined utility. As a hypothetical example to illustrate this: two models $\mathbf{M}_i, \mathbf{M}_j$ from vendors $i, j$ respectively, each have values $0.5$, then if the model seller $i$ decides to fraudulently duplicate model $\mathbf{M}_i$ to another $\mathbf{M}_{i_{\mathrm{c}}}$. Although the value for $\mathbf{M}_i$ depreciates, such duplication can lead to a higher value for vendor $i$, namely the value of $\mathbf{M}_i$ and $\mathbf{M}_{i_{\mathrm{c}}}$ combined might be higher than if only $\mathbf{M}_i$ is present.

We note that our proposed approach can be adapted to address this issue relatively easily, by substituting our proposed in Eq. (3) into the variant of the Shapley value [24, Theorem 4.5], which importantly continues to satisfy the properties (P1), (P2) and (P3) [24]. However, we highlight that (the robustness to) such duplication is beyond the scope of this work.

---

[12] Our definition of conditional redundancy is a weaker version of [24, Assumption 2] which stipulates the benefit of a copy $\mathbf{M}_{i_c}$ (conditioned on model $\mathbf{M}_i$ already being added) is exactly $0$.

# B  Additional Literature Review

## B.1  Relation to Data Valuation and its Design Approach

Data valuation, originating from the motivating application scenario of AI marketplaces [1, 13, 24], studies how to determine the intrinsic worth of data, often in the context of ML. Intuitively, as these marketplaces treat data as commodities for trading, a pricing mechanism (i.e., a valuation function) is necessary. There have been some works exploring data valuation [10, 20, 39, 80], by leveraging ML principles and assumptions. For instance, data are more valuable if training on the data produces an ML model with higher performance (i.e., more accurate). In addition to the application scenario of AI marketplaces, the value of data can also be/has been used in interpretable ML [20], data sharing [16] and collaborative ML [76, 79]. However, as motivated in Sec. 1, there are various practical scenarios where data valuation is difficult. Hence, we explore an alternative by shifting our focus onto the ML models in these scenarios to consider model valuation.

In terms of the design approach, although the existing data valuation works have different technical perspectives and thus different solutions, they often adopt a common first-principle approach, i.e., *data which can produce more accurate models are more valuable*. This approach provides an interpretation for the valuation function or the values assigned to the data. While it may seem counter-intuitive to market economics where the price/value can naturally arise from the demand and supply, this approach is sensible because the lack of effective demand (i.e., the willingness and ability of buyers to purchase goods/data/ML model at different prices). To elaborate, effective demand requires the buyers to have some intrinsic valuation function for the data/ML model which helps determine the quantity the buyers are willing to purchase at some fixed price. In contrast to the more conventional goods, the market for data/ML model is relatively niche in the sense that even the buyers themselves do not already have a good intrinsic valuation function for the data/ML model. As a result, the buyers are unable to specify the effective demand, which makes it difficult for the price/valuation to arise naturally from the demand and supply in a market. To this end, the (proposed) valuation methods (i.e., existing data valuation methods and the model valuation method in this paper) aim to fill in this gap by explicitly designing such a valuation function where the value is determined through the utility of the data/ML model in the context of ML (e.g., predictive performance). In this vein, the existing data valuation methods and our proposed model valuation method share a common perspective of designing the valuation function reflect the utility of the data/ML model in terms of some performance in the ML context. An added benefit of this design approach is the interpretability of the value. To see this, suppose in the marketplace (e.g., AWS marketplace), an auditor questions the basis for certain pricing of some ML model, our proposed valuation function can provide some insight to that question and can potentially be used by regulators to oversee the ML model marketplaces.

In terms of the practical setting, data valuation can be viewed as (mostly) white-box (i.e., the actual data are used as the input to the designed valuation function and thus completely observed).[13] Intuitively, in order to determine the value of some data, the valuation function must "see" the data. In this regard, this setting for data valuation leads to a relatively straightforward formal representation of the data, which is the data. In contrast, model valuation can encounter additional practical difficulties which make the formal representation of ML much less straightforward. Similar to in data valuation, we might want to use the model itself as its formal representation (e.g., the parameters of the parametric models). However, the so-called black-box access which is particularly appealing in model valuation (Sec. 1), excludes the choice of using the model itself (e.g., the parameters). Then, it becomes unclear what the formal representation of a model to use under this black-box access setting. In other words, for model valuation, precisely what is the input to the valuation function?

To briefly summarize the comparison between data and model valuation: In light of the practical obstacles of applying data valuation, we explore the alternative of model valuation. We adopt a similar design approach to those adopted by existing data valuation works to explicitly design a model valuation function that reflects the utility of the ML model/data in terms of a performance in the ML context. To address the additional challenges due to the black-box access setting (which is not encountered in data valuation), we propose a novel formal representation of an ML model (for classification).

---

[13]Although there is some preliminary work on using some noisy version of data for valuation [77], so it is not completely white-box, in general there is some form of access to the data or its statistic for valuation.

### B.2 Model Evaluation Criteria and Model Valuation

**Model evaluation criteria.** The value of an ML model depends on its utility/performance in the ML context. But the performance is a multi-faceted concept because there are different evaluation criteria, motivated by and suitable in different scenarios. For instance, one of the most commonly used evaluation criteria is the predictive accuracy (i.e., the proportion of correct predictions of the model). Another useful criterion is the predictive certainty (i.e., the certainty with which the model makes the predictions), as illustrated in Sec. 1. Furthermore, there are other sophisticated and practically important evaluation criteria such as fairness in prediction [56], robustness to adversaries [43] and robustness to distributional shifts in data [67]. In this regard, one approach is to design a model valuation bespoke to each of these criteria separately. However, it is more scalable and appealing to leverage a common theoretical connection among these criteria to design a more general model valuation that can be specified to different criteria as needed.

**Model valuation.** The work of [66] investigates the binary classification setting and is not suitable for empirical comparison as we consider problems with multiple classes. The works of [10, 11, 13, 51] approach the design of a model marketplace from an economics perspective by addressing issues like arbitrage (e.g., via horizontal/vertical pricing). In contrast, we formalize the value of a model via what it has learned w.r.t. a task. In addition, the black-box access setting is appealing in a model marketplace as it accommodates different model types. Some existing methods [8, 30, 33] focus on how to learn a fused model from several trained models with black-box access instead of how to value these models.

## C  Additional Experiments

### C.1  Additional Experiment Settings

**Licenses of used datasets and computational resources.**  MNIST [45]: Creative Commons Attribution-Share Alike 3.0. CIFAR-10 [41]: The MIT License (MIT). MedNIST [58]: Apache License 2.0. DrugRe [23]: Apache License 2.0 KDD99 [28]: Apache License 2.0.

We perform our experiments on a server with Intel(R) Xeon(R) Gold 6226R CPU @2.90GHz and four NVIDIA GeForce RTX 3080's. As in our experiments, we use the pre-trained weights for the models (where available) instead of training from scratch. This is because our method is w.r.t. trained models, instead of focusing on the training procedure. As a result, the usage of GPUs is moderate (mainly for performing inference on the trained models, typically within $1 - 2$ hours depending on the complexity of the models).

**Multiple independent training for robustness of results.** We train a particular setting multiple (i.e., 3) independent times and compare the results from different settings to ensure the robustness of results. For instance, Fig. 7 examines the effect of training data ratio on MSVs. For each particular training data ratio, we perform 3 independent training using the same model type and set of hyperparameters so that the plotted results are robust to randomness in the training (via stochastic gradient descent).

### C.2  Additional Discussion and Experiment Details for Learning MSV

**Learning approach.**  Conventionally, the model Shapley value (MSV) of each model $\mathbf{M}_i$ is calculated (for sufficiently small $N$) or approximated (for larger $N$, e.g., larger than 30). The computational complexity of exact calculation scales exponentially in the number of models (i.e., $\mathcal{O}(2^N)$), and that of approximation scales polynomial in $N$ [53], depending on the approximation requirement (i.e., a better approximation with smaller error would incur a higher computational cost). Furthermore, each of these computational complexities has to be multiplied by the number $N$ of models, since the calculation or approximation is performed for each model individually. Our learning approach aims to reduce the number of models for which calculation or approximation is performed, following the steps (i) perform calculation or approximation (e.g., Monte Carlo) for a subset (of size $Z$) of all $N$ models individually; (ii) use the obtained $Z$ model-MSV (or model-approximate MSV) pairs to fit a regression learner (e.g., Gaussian process regression); (iii) use the regression learner to predict the MSV for the remaining $(N - Z)$ models.

We highlight that this approach does *not* aim to reduce the computational complexity of the MSV of a model. Instead, this approach aims to reduce the *total* computational complexity of obtaining the

MSVs of $N$ models, by a factor of $N/Z$ (e.g., if $Z = 30$ for $N = 150$ models such as in Table 2, the total computational complexity is reduced to $1/5$th). Importantly, our learning approach is parallel to the prior and existing efforts that aim to reduce the computational complexity of the MSV of a model. It means that if a more efficient approximation (than the oft-used Monte Carlo) to the MSV of a model is proposed (in the future), it can be directly integrated with our learning approach in step (i) described above, namely replacing Monte Carlo.

**Why Gaussian process regression.** There are several reasons that we adopt the Gaussian process regression (GPR) as the specific choice for the learning approach: (i) GPR is a kernel-based method, which can exploit a suitably defined distance function between inputs. The Hellinger distance between two Dirichlet abstractions, or the $\ell_1$ distance between the observed sufficient statistics $\bar{h}_i$ of two models are both such suitable distance functions between the inputs (i.e., models). Hence, we adopt GPR to exploit the squared exponential kernel $\exp(-d(i, i')/(2\sigma^2))$ on the similarity measure between models (where the lengthscale $\sigma$ is learned). Specifically, for $\{\boldsymbol{\alpha}_i, \hat{\phi}_i\}$, $d(i, i') :=$ $d_{\mathrm{H}}(\mathbb{Q}_i, \mathbb{Q}_{i'})$, while for $\{\bar{h}_i, \hat{\phi}_i\}$, $d(i, i') := |\bar{h}_{i'} - \bar{h}_i|_1$. (ii) While the exact MSVs satisfy (P1)-(P4), the predicted MSVs are not guaranteed to satisfy these properties. Fortunately, if the predicted MSVs have a bounded error to the exact MSVs, then these properties can be approximately satisfied [82]. In particular, GPR has such an error guarantee [47]. The result [47, Theorem 3.1] requires the function to be learnt to be Lipschitz continuous (which we derive in Theorem 1), and also the kernel to be Lipschitz continuous, so we adopt the squared exponential kernel, which is Lipschitz continuous [47]. Moreover, note that the result [47, Theorem 3.1] is w.r.t. a continuous input space (i.e., a subset of $\mathbb{R}^d$ for some $d$), but the error guarantee depends on the Lipschitz continuity *only* through the metric between two inputs (i.e., $\ell_2$ norm of the input vectors). This is to say, their result can be adapted to our setting (where the input space is not continuous, but discrete): In our formulation for $\Phi$, the metric between two inputs is the Hellinger distance (i.e., $d_{\mathrm{H}}(\mathbb{Q}_i, \mathbb{Q}_{i'})$). It is thus an appealing future direction provide a formal guarantee based on these two theoretical results.

### C.3  Additional Results for MSV vs. Predictive Accuracy/Certainty

**Varying size of training data.** We use the same model type and vary the size of training data. For MNIST, we use a CNN, CIFAR-10, we use ResNet-18, for MedNIST we use a specific architecture called MedNet,[14] and for DrugRe we use a CNN for text. Fig. 7 shows more training data generally lead to (better trained models, and thus) higher MSVs. Although in some cases the MSVs are negative, it can be mitigated (if necessary) by exploiting the linearity property of MSV to linear translate all MSVs by a positive amount. For instance, if only ranking of the models is needed, then negative values are acceptable as long as the ordering is correct.

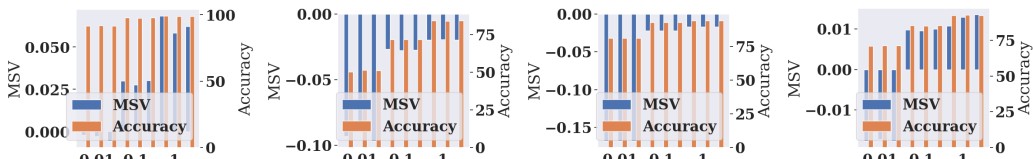

Figure 7: From left to right: MNIST, CIFAR-10, MedNIST and DrugRe. For CIFAR-10 and MedNIST, $\mathcal{D}$ is the original test set *without* partitioning and for MNIST and DrugRe we partition the original test set according to classes.

**Varying model type.** We vary the model types and train them independently on the same data. We train each model independently for 3 copies for robustness of results. For KDD99, the model types are CNN, MLP and LR. For CIFAR-10, the model types are ResNet-18, SqueezeNet and DenseNet-121. For MedNIST, the model types are ResNet-18, MedNet, and a tiny CNN. For DrugRe, the model types are CNN for text and BiLSTM. Note that these are the model types used in the empirical verification of the generalized symmetry result in Sec. 4.1.

Fig. 8 shows the following. For KDD99, MLP performs the worst (very negative MSVs) while LR performs the best. For CIFAR-10, DenseNet-121 outperforms the rest while SqueezeNet performs

---

[14]`https://github.com/apolanco3225/Medical-MNIST-Classification/blob/master/MedNIST.ipynb`

the worst. For MedNIST, the bespoke MedNet performs the best while the tiny CNN performs the worst. For DrugRe, BiLSTM outperforms CNN for text.

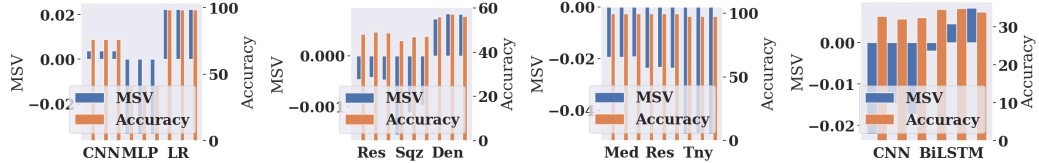

Figure 8: From left to right: KDD99, CIFAR-10, MedNIST and DrugRe. For KDD, MedNIST and DrugRe $\mathcal{D}$ is the original test set without partitioning and for CIFAR-10 we specifically use the misclassified input data from the original test set and perform partitioning according to classes.

**Varying predictive certainty.** We use the same model type (trained on the same data) and only vary the predictive certainty. The model type is LR for KDD, DenseNet for CIFAR-10, MedNet for MedNIST and CNN (for text) for DrugRe.

Fig. 9 shows increasing predictive certainty (while maintaining the predictive accuracy) improves MSVs.

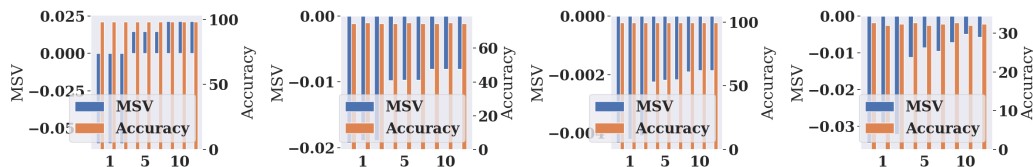

Figure 9: From left to right: KDD99, CIFAR-10, MedNIST and DrugRe. For all 4 datasets $\mathcal{D}$ is the original test set without partitioning.

## C.4 Additional Results for Identifying Valuable Models for a Larger Learner

We perform additional experiments on CovType [5], MNIST and CIFAR-100. For CovType, the models used are decision trees of depths at most 3 and the larger learner is a random forest. For MNIST (CIFAR-100) the models used are LeNet-5 [46] (Resnet-18) and the larger learner is a voting classifier. The total number of models is 50 for CovType and CIFAR-10, and 25 for CIFAR-100.

Fig. 10 shows for all three datasets, MSVs effectively identify the valuable subset of models since following the highest-first sequence increases the test accuracy more quickly. In particular, we identify overfitting for CovType and MNIST since the max attained accuracy occurs before all the models are included.

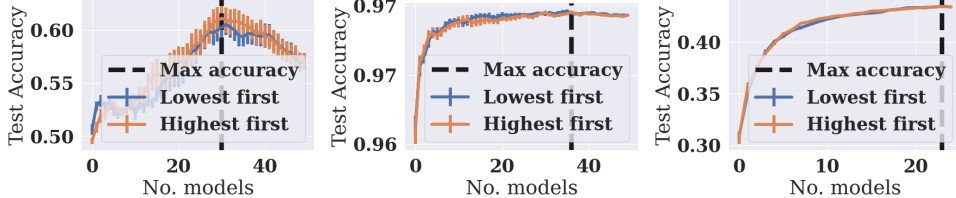

Figure 10: From left to right: CovType, MNIST, and CIFAR-100. Test accuracy vs. number of included models.

## C.5 Empirical Advantage of the Chernoff Distance over the Hellinger Distance

**Additional performance metrics of learning MSV.** While using $d_{\mathrm{H}}$ produces high regression performance of learning MSV (i.e., low test errors), we find that the performance may be somewhat limiting when we apply other performance metrics such as coefficient of determinant ($R^2$) and

explained variance (exVar). In particular, we find thet $d_C$ outperforms $d_H$ on these two metrics, shown in Table 4.

Table 4: Average (standard errors) of test performance (over 10 random trials) on $80\%$ of data after training on $20\%$ of data for MNIST (top two rows) and CIFAR-10 (bottom two rows). Left (right) two columns correspond to using $d_H$ ($d_C$). For both metrics, higher is better (optimal is 1).

|  | $R^2$ | ExVar | $R^2$ | ExVar |
|---|---|---|---|---|
| $\boldsymbol{\alpha}_i$ | $0.29(6.72e^{-3})$ | $0.201(2.90e^{-2})$ | $\mathbf{0.93}(2e^{-3})$ | $\mathbf{0.92}(3e^{-3})$ |
| $\bar{h}_i$ | $-4.04(0.462)$ | $-3.66e^1(1.33)$ | $\mathbf{0.85}(7e^{-3})$ | $\mathbf{0.44}(3e^{-2})$ |
| $\boldsymbol{\alpha}_i$ | $0.73(2.00e^{-2})$ | $0.67(3.76e^{-2})$ | $\mathbf{0.93}(2e^{-2})$ | $\mathbf{0.91}(3e^{-2})$ |
| $\bar{h}_i$ | $-8.69(1.21)$ | $-2.20e^1(2.47)$ | $\mathbf{0.80}(1e^{-2})$ | $\mathbf{0.76}(2e^{-2})$ |

Notice in Table 4, the right two columns ($d_C$) outperform the left two columns ($d_H$). We hypothesize that this can to the logarithmic dependence in $d_C$ resulting a more "linear" relationship between $\mathbb{Q}_i$ and $\phi_i$. For Dirichlet distributions (i.e., Dirichlet abstractions) over a $C$-dimensional space ($C > 1$), the product of the pdf (appears in both $d_C$ and $d_H$) over the space may not be well-behaved, and the integral of this product makes it more intractable (possibly due to the curse of dimensionality). The logarithmic operation in $d_C$ can help mitigate this, resulting in the better regression performance using GPR, in terms of exVar and $R^2$.

## C.6 Additional Benefit of A Lower Level of Dirichlet Abstraction

Recall the example in Sec. 2 that using a lower level of Dirichlet abstraction by partitioning $\mathcal{D}$ according to the classes allows us to correctly differentiate $\mathbf{M}_i$ from its 'shifted' version $\mathbf{M}_{i'}$. Intuitively, partitioning $\mathcal{D}$ according to the classes improves the similarity measure between $\mathbf{M}_i$ and $\mathbf{M}_{i'}$ (via some distributional distance) and we hypothesize that it can also improve the generalized symmetry, which exploits the similarity measure between two models (e.g., via $d_C(\mathbb{Q}_i, \mathbb{Q}_{i'})$). We verify this on MNIST and CIFAR-10 by partitioning the respective query sets $\mathcal{D}$ (i.e., test set) according to $C = 10$ classes s.t. query set $\mathcal{D}_k$ is from class $k$ and $\gamma_k := |\mathcal{D}_k|$. We compute $d_C(\mathbb{Q}_i, \mathbb{Q}_{i'}; \{\mathcal{D}_k\}_{k=1,...,C}) := \sum_{k=1}^{C} \gamma_k \, d_C(\mathbb{Q}_{i,\mathcal{D}_k}, \mathbb{Q}_{i',\mathcal{D}_k})$ and apply (P3) to compute $\phi_i(\{\mathcal{D}_k\}_{k=1,...,C}) = \sum_{k=1}^{C} \gamma_k \, \phi_i(\mathcal{D}_k)$, as shown in the right two plots of Fig. 6. Note that we use the Chernoff distance (instead of the Hellinger distance) due to its numerical stability, and will elaborate later on this point.

The setting for this experiment is as follows, we utilize the MNIST and CIFAR-10 datasets respectively. For each dataset, we train $N = 150$ models with 3 different model types (50 of each). Specifically, for MNIST, we train 50 of LR, MLP and CNN while for CIFAR-10, we train 50 of ResNet-18, SqueezeNet and DenseNet-121. As in Sec. 4.1, we evaluate the performance via the Pearson correlation coefficient between $d_C(\mathbb{Q}_i, \mathbb{Q}_{i'}; \{\mathcal{D}_k\}_{k=1,...,C})$ and $|\phi_i(\{\mathcal{D}_k\}_{k=1,...,C}) - \phi_{i'}(\{\mathcal{D}_k\}_{k=1,...,C})|$ in Table 5.

Table 5: Comparison of Pearson coefficients between using single query set $\mathcal{D}$ vs. partitioned query sets $\{\mathcal{D}_k\}$ for MNIST (top two rows) and CIFAR-10 (bottom two rows).

| Pearson $\backslash N$ | 60 | 90 | 120 | 150 |
|---|---|---|---|---|
| $\phi_i(\mathcal{D})$ | **0.9493** | **0.9378** | 0.9146 | 0.9011 |
| $\phi_i(\{\mathcal{D}_k\})$ | 0.9440 | 0.9362 | **0.9326** | **0.9274** |
| $\phi_i(\mathcal{D})$ | 0.9958 | **0.9955** | 0.9934 | 0.9898 |
| $\phi_i(\{\mathcal{D}_k\})$ | **0.9961** | 0.9949 | **0.9936** | **0.9924** |

Results in Table 5 confirm our hypothesis that a lower level of Dirichlet abstraction can lead to better observation of the symmetry result. This is because a lower level of Dirichlet abstraction provides a more refined representation of each model (i.e., w.r.t. individual classes instead of overall). In particular, we note the performance improvement for a larger $N$ is important so that model Shapley is learnable to be adopted in a large-scale marketplace.

**Numerical overflow prevents the Hellinger distance from performing well.** Due to the numerical overflow issue of $d_{\mathrm{H}}$: If the argument to $\Gamma(\cdot)$ (e.g., the denominator in $B(\boldsymbol{\alpha})$ in Definition 1) approaches 171.614479 in double precision floating point numbers, from [63]. Recall Definition 3 fuses models in a coalition $\mathcal{C} \subseteq [N]$ to obtain $\mathbb{Q}_{\mathcal{C}} = \mathrm{Dir}(\sum_{i=1}^{n} \alpha_{i,1}, \ldots, \sum_{i=1}^{n} \alpha_{i,C})$ where $|\mathcal{C}| = n$. The summation $\sum_{i=1}^{n} \alpha_{i,1}$ can lead to overflow when $n$ (i.e., $|\mathcal{C}|$) is large, which tends to happen when $N$ (the total number of models is large) because the calculation of MSV $\phi_i$ requires the enumeration of $\mathcal{C} \subseteq [N] \setminus \{i\}$ to obtain the respective $\mathbb{Q}_{\mathcal{C}}$. For instance $N = 150$, then the largest $\mathcal{C} \subseteq [N] \setminus \{i\}$ contains 149 models and if their respective parameter $\alpha_{i',k}$ for the class $k$ is on average larger than $171.614479/149 \approx 1.15$, then it leads to numerical overflow for $\Gamma(\cdot)$ (and thus Hellinger distance). Note that the Chernoff distance avoids this by combining the logarithmic operation with the $\Gamma(\cdot)$: in implementation $\ln \Gamma(\cdot)$ can be computed directly instead of first computing $\Gamma(\cdot)$ and then the logarithmic operation.

### C.7 Same Model Types Produce Distributionally Close Dirichlet Abstractions

We perform clustering (using $d_{\mathrm{C}}$ as the distance measure instead of $d_{\mathrm{H}}$ because it shows a better visual illustration and the clusering results) to illustrate the effects of model types on the similarity between the resulting Dirichlet abstractions using the default test set as the query set without perform partitioning according to classes. Specifically, for the $N = 150$ models trained on MNIST (50 CNNs, 50 MLPs and 50 LRs), we apply the *density-based spatial clustering of applications with noise* (DBSCAN) [17], because it is suitable for data which contain clusters of similar density (in our case, 3 clusters each of size 50). Fig. 11 shows good separation of different model types, supported by additional quantitative indices: homogeneity (1.000), completeness (0.971) and V-measure (0.985). Homogeneity measures the degree to which a single cluster contains only members of a single class. Completeness measures the degree to which the all the members of a single class are correctly classified into the same cluster. V-measure is a harmonic mean of both homogeneity and completeness. These high values suggest the Dirichlet abstractions contain sufficient model information to clearly distinguish between these model types. As a result, the clustering performance is quite good via the adjusted random index (ARI) (0.990 and optimal is 1.0), which measures the correctness of the given clustering via its match with the ground truth. We consider ARI because it is suitable for large equal-sized clusters [64], which is this case here: 3 equal-sized clusters of size 50. This result confirms the effectiveness of our approach to "convert" the heterogeneous models into homogeneous Dirichlet abstractions which live in the metric space (with metric $d_{\mathrm{H}}$, though we highlight the experiments are performed w.r.t. $d_{\mathrm{C}}$ due to practical concerns). In particular, the numerical stability issue of $d_{\mathrm{H}}$ [63] prevented us from obtaining meaningful clustering results using $d_{\mathrm{H}}$. Moreover, this result motivates our grouping paradigm used in our experiments to verify the generalized symmetry.

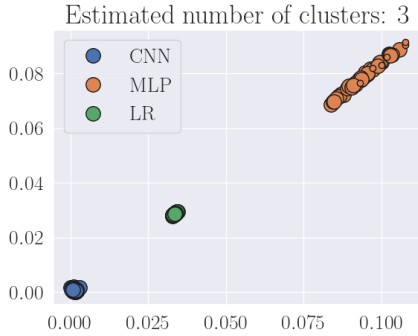

Figure 11: DBSCAN clustering of $N = 150$ models trained on MNIST based on the Chernoff distance $d_{\mathrm{C}}$ Definition 4.

