# OpenReview forum: "Model Shapley: Equitable Model Valuation with Black-box Access"
_NeurIPS.cc/2023/Conference — NeurIPS 2023 poster_

### Official Review · Reviewer_Coej · 2023-06-28

**Soundness:** 3 good
**Presentation:** 1 poor
**Contribution:** 3 good
**Rating:** 4
**Confidence:** 2

**Summary:**

This paper studies an abstract problem of model valuation.
They propose a notion of valuing models, called the *Model Shapley Value*, based on the classic notion of Shapley Value from the literature on cooperative games.
Additionally, the work proposes an abstraction for models, a *Dirichlet abstraction*, that is meant to enable efficient comparison of different models.
They propose a method of learning to valuate models, and evaluate their method empirically.

**Strengths:**

The problem of valuing different predictive models is well-motivated.  This paper nicely motivates the study of valuation of models, rather than simply data, on a number of axes.  The reliance on Shapley value is standard in the ML literature, but also well-motivated, due to the strong theoretical guarantees and study of Shapley.

**Weaknesses:**

Major concern:  I find the presentation of the paper to be a substantial barrier to understanding and evaluating the work.

Concretely, the task at hand of model valuation is never actually described formally.
- What is the task?  What is the input and output?
- What is the trivial solution?
- Why is the proposed solution better?

Unfortunately, I do not understand the description of Dirichlet abstractions.
- Again, what is the point of the abstraction?  We are representing a model as a Dirichlet probability distribution?  Why?
- What is $\mathbf{M}_i(x)$?  Specifically, what is $x$ here?  Is this the induced distribution on predictions given a randomly sampled input $x_j$?
- Why can this be viewed as a Dirchlet distribution?  I see the citation, but it feels like a substantial enough point that it should be justified and motivated in the text of this paper.
- "$\mathbb{Q}_i$ encodes the predictive accuracy and certainty of $\mathbf{M}_i$ through a theoretical connection..." << Is this a formal statement?  How does this work?  How is it justified?


My lack of understanding of the basic building blocks behind this paper makes it difficult for me to evaluate the theory or experiments appropriately.  I've spent a decent amount of time trying to understand, and it is still not clear to me what is happening.

**Questions:**

Many of my questions are listed above.
- What is the task at hand?
- What is the key goal of this paper?
- What are the key contributions to achieving this goal?

I believe that there is probably something interesting happening in this paper, but I am not able to evaluate due to my lack of clarity on the basic objects studied.

**Limitations:**

Adequately addressed.

---

> ### Author Rebuttal · Authors · 2023-08-08
>
> We thank Review Coej for reviewing our paper and for finding our problem "well-motivated" and the usage of the Shapley value also "well-motiavated".
>
> We wish to address the questions as follows.
>
> W1.
>
> > What is the task?  What is the key goal of this paper?
>
> As pointed out by the reviewer:
> > This paper nicely motivates the study of valuation of models,
>
> __The task or key goal is to study the valuation of models, and proposes a suitable method for model valuation__, requiring us to resolve three main technical challenges as in Sec. 1:
>
> - (i) how to represent a model (which we call an abstraction) such that it is amenable to valuation? (lines 44-54);
>
> - (ii) how to ensure equitability properties are satisfied, so as to ensure the fairness of a model marketplace? (lines 54-60);
>
> - (iii) in leveraging the Shaley value to satisfy these equitability properties, a computational challenge arises, how then to resolve this challenge? (lines 61 - 66).
>
> > What are the key contributions ... ?
>
> __Resolving the above three challenges constitutes the key contributions, as summarized in lines 67-86 of Sec. 1.__ To elaborate, as the reviewer pointed out, the Dirichlet abstraction is proposed to resolve (i) as a suitable model abstraction, that is amenable to valuation. Then, the Shapley value formulation is adopted via designing a specific function $\nu$ (in Equ. (3) in line 195) to satisfy these equitability properties and resolve (ii); and a learning approach, by leveraging the analytic properties of the the proposed function $\nu$ and Dirichlet abstractions, is proposed to resolve (iii).
>
> > What is the input and output?
>
> The formal problem is described in Sec. 3.1. The input is $N$ models $\\{ \mathbf{M}\_1,\ldots , \mathbf{M}\_i, \ldots , \mathbf{M}\_{N} \\}$ with only black-box access.
>
> The desired output consists of  __the definition of $\phi\_i$ for each model, which satisfies the equitability properties P1-P4__ (lines 181-184) and  __a computationally feasible implemetation to obtain $\phi\_i$__.
>
> > What is the trivial solution?
>
> __A trivial solution is to use the predictive accuracy of a model as its value (lines 47-48)__.
>
> > Why is the proposed solution better?
>
> __Lines 48-53 highlight why the trivial solution is too reductive and thus missing other key aspects of the model__, such as its predictive certainty. Our proposed solution is shown to be consistent with both the predictive accuracy (Figure 3 left) and certainty (Figure 3 right). Moreover, our proposed solution is extendable to more sophisticated evaluation criteria (Sec. 3.2 and Table 1).
>
> W2.
>
> > Again, what is the point of the abstraction?
>
> __The abstraction is meant to be formal representation of the model__. Recall that we consider the black-box access setting (motivated in lines 33-38), which means only queries and the model predictions are observed. In this way, without a formal representation, it is difficult to provide a formal treatment of the problem of model valuation.
>
> > We are representing a model as a Dirichlet probability distribution? Why?
>
> As pointed out by the reviewer, "Additionally, the work proposes an abstraction for models, a Dirichlet abstraction, that is meant to enable efficient comparison of different models."
> There are two main reasons: (i) __to enable efficient comparison__ by using the closed-form expressions of the Hellinger distance between Dirichlet distributions; (ii) __to provide theoretical analysis__ by leveraging the analytic properties of the Hellinger distance between Dirichlet distributions, as in Proposition 2 and Theorem 1.
>
> > What is $\mathbf{M}_i(x)$?
>
> A model $\mathbf{M}\_i$ is a mapping, namely $\mathbf{M}\_i: \mathcal{X} \mapsto \triangle(C)$.
>
> > Specifically, what is $x$ here? Is this the induced distribution on predictions given a randomly sampled input $x_j$?
>
> $x \in \mathcal{X}$ is a specific feature vector, viewed a a realization of a random variable $X \sim P_X$ whose $\text{supp}(X) = \mathcal{X}$ (and $P_X$ is represented empirically by the task). Then, $\mathbf{M}\_i(X)$ is an induced distribution (by $P_X$) over $\triangle(C)$.
>
> > Why can this be viewed as a Dirchlet distribution?  ...
>
> Because (i) __the support of $\mathbf{M}\_i(X)$ matches _exactly_ to that of a Dirichlet distribution__; (ii) from a statistical viewpoint, __the Dirichlet distribution is a suitable modeling choice for the distribution $\mathbf{M}\_i(X)$ [57]__. We will make this point explicit in our revision.
>
> > "$\mathbb{Q}_i$ encodes the predictive accuracy and certainty of $\mathbf{M}_i$ through a theoretical connection..." << Is this a formal statement? How does this work? How is it justified?
>
> __The formal result is Proposition 2__. The insight is that the cross entropy (CE) loss of a model encodes the predictive accuracy and certainty (see below). The CE loss is used to construct upper and lower bounds for our proposed method (i.e., $\nu$ in Equ.(3)). Hence, our proposed method also encodes the predictive accuracy and certainty.
>
> Recall the CE loss of a $C$-dimensional predicted probability vector $\hat{y}$ w.r.t. the one-hot encoded true label $y$: $-\sum_{k=1}^C y_k \times \ln(\hat{y}_k) $.
>
> W.l.o.g., assume that $y_{1}=1$ (i.e., the correct class is the first class).
>
> - For two predictions $[0.9,0.1, 0,\ldots,0]$ vs.$[0.1, 0.9,0, \ldots, 0]$. The CE losses are $0.105$ and $2.30$, respectively. Note that the first prediction is correct while the second in incorrect and that both predictions are "equally certain". Hence, __A higher predictive accuacy implies a lower CE__.
>
> - For two predictions $[0.9,0.1, 0,\ldots,0]$ vs.$[0.6, 0.4,0, \ldots, 0]$.  The CE losses are $0.105$ and $0.511$, respectively. Note that both predictions are correct while the first prediction is "more certain". Hence, __A higher predictive certainty implies a lower CE__, if the prediction is correct.
>
> We hope our clarifications have addressed your questions and helped improve your opinon of our work.

---

> > ### Comment · Reviewer_Coej · 2023-08-13
> > **Receipt of rebuttal**
> >
> > I continue to find the presentation very confusing.  I recognize that other reviewers seem to be more confident in their understanding of the paper, so I will downweight my confidence and make my score less extreme.  Regardless of outcome, I recommend that the authors work on outlining the problem at hand precisely, formally, and to distinguish between the *problem* (i.e., given models, what are the values of them?) and solution concepts (i.e., equitable Shapley values, using ML to predict model values, etc).
> >
> >
> > ***
> >
> > Based on more time with the manuscript and others' comments, I will update my review.
> > It is now clear to me that there is something interesting happening here, with the representation of the models, and in particular, in connecting the idea of closeness in Hellinger distance with the ability to learn MSVs.
> >
> > For me, however, this took a lot of work to understand.
> > I think there are a few things that were not obvious to me, and would be worth belaboring in the intro or early preliminaries.
> >
> > - I would lay out, a bit more directly, the tasks at hand:
> > (a)  Given n models, produces a valuation for each
> > (b)  Given a sample of models, produce a valuation function (i.e., model appraiser) that on an unseen model, returns a valuation.
> >
> > - I think some additional specificity about what type of "models" and "tasks" you're thinking about would be helpful orientation. For instance, I'm not sure I understand the conflation of "task" and "query set".
> >
> > - Finally, I think the paper is very notationally dense, but it is worth spelling out very clearly the notation and terminology being used, and to define it before you use it.  For instance, Q* is defined inline after it's already used.
> >
> >
> > On a technical level, Theorem 1 is interesting.  I think the presentation would improve even more if there was more exposition about how to connect the theoretical guarantees of Theorem 1 to the problem of learning MSVs.  In what sense does Theorem 1 provide guarantees for the learning task?  In what sense does it give a guiding principle that, in empirical evaluations, turns out to be wise?

---

> > > ### Author Response · Authors · 2023-08-15
> > > **Thank the reviewer for the additional time and raising the score**
> > >
> > > We thank Reviewer Coej for the taking the extra time and effort, we really appreciate it!
> > >
> > > We thank the reviewer for the suggestions on writing and, in our revision, we will
> > > - clearly lay out the formal problem statement in terms of the inputs (i.e., $N$ classification models) and desired outputs (i.e., an equitable valuation function $\phi_i$ and a computationally efficient way to obtain it such as via a learnt appraiser);
> > >
> > > - include additional descriptions of the models and tasks w.r.t. our problem setting: the models are trained classification models with specified input and output spaces (without constraints on their architectures) and the tasks are the classification problem (with the same input and output specifications as the models) that these models are trained on;
> > >
> > > - spell out clearly the list of notations (in Appendix) and ensure that $\mathcal{Q}^*$ is defined before it is formally used.
> > >
> > > [Regarding the "task" and "query set"]: The **"task"** (mentioned in line 44) **is a conceptual definition used to describe what the model is used for (by the user)**. An example would be the classification of MNIST digits. The **"query set"** (mentioned in line 39) **is meant to be how a task is formally represented**. In the example where the task is classification of MNIST digits, the corresponding query set can be a validation set containing the MNIST digit images and the labels. Hence, a "task" is the conceptual definition whose formal representation is the "query set", so in our writing we use them interchangebaly. Note that however, in our formal treatment, the query set is made precise as $\mathcal{D}$ (or the class-specific $\mathcal{D}_k$).
> > >
> > > [Regarding Theorem 1] We thank the reviewer for finding that it is interesting and for the insightful questions regarding the implications of Theorem 1, and wish to provide the following clarifications.
> > >
> > > > In what sense does Theorem 1 provide guarantees for the learning task?
> > >
> > > The main point of Theorem 1 is to show that the **model Shapley as a function, is learnable due to its "Lipschitz continuity"**; and because it is learnable, we propose to learn it.
> > >
> > > To see the Lipschitz continuity of model Shapley as a function, recall the input to this function is a model $\mathbf{M}\_i$ formally as its Dirichlet abstraction $\mathcal{Q}\_i$; and the corresponding output is its MSV $\phi_i$. Recall that the Lipschitz continuity of a function states that the difference in the outputs is bounded by (a Lipschitz constant times) the difference in the inputs, which is precisely what (the right hand side of the implication of) Theorem 1 aims to provide:
> > > $ |\phi\_i - \phi\_{i'}| \leq d\_{\text{H}}(\mathcal{Q}\_i, \mathcal{Q}\_{i'}) $
> > > where _absolute difference in the MSVs is the difference in outputs_ whilst _the Hellinger distance between the Dirichlet abstractions is the difference in the inputs_.
> > >
> > > > In what sense does it give a guiding principle that, in empirical evaluations, turns out to be wise?
> > >
> > > To exploit this Lipschitz continuity, we propose **a learning approach using Gaussian process regression (GPR) as the learner** where the specific choice of GPR is due to **a uniform regression error bound for Lipschitz continuous functions** [40].
> > >
> > > Then, in terms of implementation choice and empirical evaluations, GPR requires a kernel (between inputs) defined w.r.t. some suitable distance between the inputs, so **the Hellinger distance** presents as a very natural choice (lines 890-896 in App. C2), which **proves to be an effective choice**, demonstrated via the low regression errors in Fig. 3.
> > >
> > > We hope our clarifications help provide additional exposition about how to connect the theoretical guarantees of Theorem 1 to the problem of learning MSV. We are happy to clarify further questions. Again, we thank Reviewer coej for the feedback and questions.

---

### Official Review · Reviewer_bude · 2023-07-04

**Soundness:** 3 good
**Presentation:** 3 good
**Contribution:** 2 fair
**Rating:** 6
**Confidence:** 3

**Summary:**


The authors represent the model's predictive accuracy and certainty with Dritchlet abstractions and formalize model Shapley values, which measure the value of models to a given task defined by a query set.
They evaluate their work on MNIST, CIFA-10, DrugRe, and medNIST datasets.

**Strengths:**

- The authors lay out an interesting problem, with clear background and supporting statements.

- The authors formulate a solid, well-thought-out solution to the problem.

- The theoretical statements are sound, and the results of the empirical analyses are clearly explained.

- The paper is well structured and written, thus facilitating a good read.



**Weaknesses:**


P4: Diminished marginal utility
- While it might be the case that the marginal utility of duplicate models reduces, the model sellers might negatively affect the marketplace with untruthful model presentations. Say, for example, two models (A, B), each have values 0.5, then if one model seller decides to fraudulently duplicate model A, the value although depreciates, might assign a high value collectively to A (2/3).

Query set
- The authors say they have black-box access to the models. This is somewhat okay with image data, where one doesn't need explicit knowledge of the key features. For example, if I'm interested in testing the value of cat/dog classification models, I will bring cat/do images in the query set. However, for things like loan worthiness, with black box access, how does one construct the query set?
- What is the upper and lower bound on the query set size before one reverse engineer the model?

P3
- Multiple complementary tasks might be a fair way to measure value of models with p3 assumption. How about if these tasks are tradeoffs of each other?




**Questions:**


For the rebuttal, the authors should respond to all the questions highlighted in the weaknesses subsection.

**Limitations:**


The authors do not mention any limitations of their work.

---

> ### Author Rebuttal · Authors · 2023-08-08
>
> We thank Reviewer bude for taking the time to review our paper, and for appreciating our studied problem ("lay out an interesting problem"), proposed approach ("a solid, well-thought-out solution"), presented results ("theoretical statements are sound, and the results of the empirical analyses are clearly explained") and our writing ("paper is well structured and written, thus facilitating a good read"). We wish to address the questions as follows.
>
> W1.
> > P4: Diminished marginal utility
>
> __Our proposed approach can be adapted to address this issue relatively easily__, by substituting our proposed $\nu$ in Equ.(3) into the variant of the Shapley value [Theorem 4.5, 21], which importantly continues to satisfy the properties P1, P2 and P3 [21]. However, we wish to highlight that (the robustness to) such duplication is beyond the scope of this work and we will include this discussion in our revision.
>
>
> W2.
> > However, for things like loan worthiness, with black box access, how does one construct the query set?
>
> __Formally, there is no difference between such different types of data__, as we have also considered non-image data, specifically tabular data (i.e., KDD99) and text-based data (i.e., DrugRe) in our experiments.
> The key requirement is that the __input and output specificaions of the model, have to match the input and output specifications of the collected data in the query set__. We believe this is a reasonable practical requirement because (i) without knowing the correct input of a model, a user cannot use the model for predictions; (ii) the output is $C$-dimensional for a $C$-way classification model.
>
> To illustrate, for an image task, the black-box model (which could be logistic regression or a convolutional neural network), takes as input an image and returns as output a predicted probability vector. Similarly for other forms of data (e.g., tabular), the model takes as input a data point (i.e., a row in the tabular data) and returns as output a predicted probability vector.
>
> > What is the upper and lower bound on the query set size before one reverse engineer the model?
>
> To the best of our knowledge, no known theoretical bounds are available to completely reverse engineer the model _only_ based on quries; __our approach is empirically shown to work with query sets which are smaller than an existing reverse-enginnering attack that requires additional assumptions _not_ satisfied in our setting__. Indeed, this is a main motivation for adopting the black-box (access) model in our setting as in lines 36-37.
>
> To elaborate, (Oh et al., 2018) requires (i) knowing something about the model (to be reverse engineered), such as "a diverse set of white-box models ... that are expected to be similar to the target black box at least to a certain extent"; (ii) hundreds to thousands of queries, to construct a model that _only predicts similarly_, __not__ the black-box model itself. In our setting, (i) is not satisfied; moreover, in our experiments, we find that query sets of size as small as $100$ are sufficient, which is much smaller than that used in (Oh et al., 2018).
>
> W3.
>
> > Multiple complementary tasks might be a fair way to measure value of models with p3 assumption. How about if these tasks are tradeoffs of each other?
>
> __If the user knows the importance of these tasks, then the user can specify the weights to achieve a desirable tradeoff.__ This is because different users might have different preferences and there is no one-size-fits-all solution. To elaborate, suppose the user _only_ cares about whether the model makes accurate predictions but not at all about adversarial robustness because the user intends to deploy it in a controlled and safe environment, then the task constructed for adversarial robustness is not very relevant to this user. In contrast, if the user does care about the adversarial robustness (which, is often at trade-off against pure predictive performance), then the user can set the weights between the two tasks according to their preferences.
>
> On the other hand, if the tradeoffs of the tasks are unknown, for instance the objectives are very complex, then uncovering the relationship between tasks (which are potentially tradeoffs of each other) is useful. Specifically, the approach to obtain the connections in Table 1 is useful. For instance, predictive accuracy and adversarial robustness are tradeoffs of each other since the objective of adversarial robustness "balances" between the clean and adversarial cross entropy (CE) losses. Upon identifying this theoretical connection, the user can then specify the weight between the two accordingly. We will include this discussion in our revision.
>
> *References.*
> Oh, Seong Joon and Augustin, Max and Schiele, Bernt and Fritz, Mario. Towards Reverse-Engineering Black-Box Neural Networks. In ICLR, 2018.
>
> We thank Review bude for the positive feedback and the comments. We hope that our response has clarified the questions, and has helped raise your opinion of our work.

---

> > ### Comment · Reviewer_bude · 2023-08-13
> >
> > I  thank the authors for investing lots of effort in answering our queries.
> >
> > In general, I find the responses reasonable and agree with most of them.  Below are some responses I didn't fully agree with;
> >
> > Query sets and Blackbox access;
> > - I still find the black-box access explanation for tabular datasets insufficient, especially in practical settings. Say, for example, a school admission system. While there is a noisy idea of what schools consider for admission, it's unclear what features are used in admissions model training. Additionally, these vary across schools. So if one wanted to valuate admission models with black-box access, query sets might be flawed or significantly vary across models, which beats the design.
> >
> > - A classification model (e.g., CV model) focusing on the image background instead of a holistic image is a model limitation, not a characteristic. On the other hand, some tabular models leaving out some features is an actual design, not necessarily a flaw. A tabular model not considering causal relationships and other relationships between features and classes would be a model design flaw. Consideration of different features is characteristically different models, not model variants.
> >
> > - In my opinion, letting 'outsiders' (e.g., not the target school admin, but say another school's admin) know what features are being used in a tabular model makes access "grey" (partial access) and not "black" (no access)  or "white" (full access).
> >
> >
> >
> > Task tradeoffs
> > - I agree that different users might have different preferences, and there is no one-size-fits-all solution. However, I think my question was more towards the divergence of this knowledge between model sellers and if this could encourage gaming. I agree that some tasks might be niche, and therefore if a buyer needed models that do that particular task, they would seek out the valuation of models for that task. However, if one seller had prior knowledge that the buyer cared about task-x+accuracy and the other sellers didn't and focused on only accuracy, this asymmetry of information in the marketplace would cause instability. So it might be better to mention this as a design limitation.
> >
> > I appreciate the responses, and I agree with some of them. Even though I didn't fully agree with everything, I think the authors' responses are generally reasonable and well thought out. I, therefore, will raise my score to 6.

---

> > > ### Author Response · Authors · 2023-08-15
> > > **Thank the reviewer for raising the score**
> > >
> > > We thank Reviewer bude for the quick response, providing constructive feedback, and raising the score.
> > >
> > > We would like to provide some discussion as follows,
> > >
> > > [On Query sets and Blackbox access] We thank Reviewer bude for raising the questions (e.g., different feature spaces, and the access to the actual feature space) for the practical application scenarios of our studied problem setting.
> > > We believe these are interesting and very relevant questions worth careful explorations. In this regard, we wish to point out that our paper aims to take _a first step towards a theoretical framework in which these questions can receive a formal treatment_ and we definitely hope to inspire other works in this direction to tackle these interesting and relevant questions. We will highlight these questions as future work in our revision.
> > >
> > > [On task tradeoffs]
> > > We think that the asymmetry of information does exist in general marketplaces (i.e., not restricted to for data or machine learning models) where the party (i.e., either buyer or seller) with more information would have an advantage. Nevertheless, we thank the reviewer for the astute observation on a concrete example of this in model marketplaces and will definitely include this point in our revision!
> > >
> > > Again, we really appreciate the reviewer's effort and thought in reviewing work and response, and providing constructive feedback.

---

### Official Review · Reviewer_RTGV · 2023-07-06

**Soundness:** 3 good
**Presentation:** 3 good
**Contribution:** 2 fair
**Rating:** 4
**Confidence:** 3

**Summary:**

This paper introduces a novel approach to model comparison and valuation, utilizing a method known as Dirichlet Abstraction. The fundamental idea is to abstract the predictive behavior of different models via a Dirichlet distribution. This abstraction allows the comparison of diverse models on an equal footing.

Three main challenges in model valuation are identified in the paper: developing a suitable abstraction for model valuation relative to a task, satisfying equitability properties in model valuation, and exploiting the equitability properties of the Shapley value in a large marketplace.

To address these challenges, the authors introduce an innovative approach to model valuation, utilizing a Dirichlet distribution to approximate a model’s predictive pattern or distribution with respect to a task. This abstraction, termed the Dirichlet abstraction, incorporates both the model's predictive accuracy and certainty. The authors then propose using the model Shapley value as an equitable valuation method, leveraging the Dirichlet abstractions' ability to preserve similarity between models. To address the computational challenge of Shapley value in a large marketplace, the paper suggests a learning approach for training a model appraiser. This model appraiser, trained on a small subset of models and their model Shapley values (MSVs), can predict other models’ MSVs, thus validating model Shapley’s practical feasibility in a large-scale marketplace.

The paper's empirical validation, performed on real-world datasets and with up to 150 heterogeneous models, confirms that higher predictive accuracy, more suitable model types, and higher predictive certainty correlate with higher model Shapley values. The authors also provide a use case for identifying a valuable subset of models from the marketplace to construct a more complex learner.

**Strengths:**

The paper presents a novel approach to model comparison and valuation using Dirichlet abstraction, offering a creative combination of existing statistical concepts applied in a new manner. The concept of utilizing Dirichlet distributions to abstract predictive behaviors of models is quite innovative. Furthermore, the introduction of class-specific Dirichlet abstraction is an original refinement to their method, preserving more detailed information about the model's behavior.

The quality of the research seems to be high, given the soundness of the mathematical framework used in the paper and the logical structure of the presented methodology. The authors provide a theoretical background to support their claims and supplement it with visual aids, offering a detailed explanation of how their method works. The decision to use Hellinger distance to measure the dissimilarity between two probability distributions shows a thoughtful choice in ensuring computational efficiency.

The ability to compare different types of machine learning models in a unified framework is an important advancement in the field. This has potential implications for model selection in various machine learning tasks, enabling a more flexible, efficient, and fair comparison of models. Moreover, the efficient computation of the Hellinger distance between Dirichlet distributions has implications in other fields where these statistical concepts are used. Overall, the paper presents a valuable addition to the machine learning and statistical literature.

**Weaknesses:**

While the paper does present a novel approach to model valuation, there are several areas where it could be improved.

The paper assumes the availability of a large query set for accurate MLE estimation. It should address situations where the available data is sparse or not balanced across classes. A detailed discussion or a possible solution for handling these scenarios would strengthen the paper.

While the authors discuss the trade-off between the abstraction level vs. query set size, there is a lack of clear guidelines or a framework for determining the optimal trade-off. This could lead to difficulties in implementing the proposed method in real-world applications. Adding a more formal discussion or a proposed methodology to address this trade-off would be beneficial.

The approach assumes homogeneity across models after the Dirichlet abstraction, which might not always be the case in practical scenarios. Some models might have specific characteristics that cannot be captured by the Dirichlet distribution. More discussion on how such cases can be handled would improve the paper.

The paper lacks a comparison with other existing model valuation methods. It would be beneficial to include an evaluation of how the proposed method fares against existing methods in terms of accuracy, efficiency, and robustness.



**Questions:**

How does the proposed technique compare with existing model valuation techniques, in terms of accuracy, computational efficiency, and other relevant metrics? It would be beneficial if the authors could provide a comparative study in this regard.

How does the method perform when the available data is sparse or unbalanced across classes? Could the authors elaborate on how their approach handles diverse model architectures, especially those with characteristics that cannot be adequately captured by a Dirichlet distribution?

Could the authors elaborate on how to choose an optimal trade-off between the level of Dirichlet abstraction and the size of the query set? A concrete set of guidelines or a decision-making framework would be helpful for practitioners seeking to apply this method.

Some of the assumptions in the paper (like model homogeneity post-abstraction) might not hold in all practical scenarios. Could the authors elaborate on the implications if these assumptions are violated and how such situations could be handled?

**Limitations:**

The authors have not explicitly addressed the potential limitations and broader societal impacts of their work.

The authors could provide more discussion on the potential limitations of their work. For instance, they might discuss the sensitivity of the model's valuation to the choice of the task or query set, and the limitations of the Dirichlet abstraction in approximating a model's predictive pattern or distribution.

In terms of broader societal impacts, the authors might consider discussing how their proposed model valuation framework could potentially affect the machine learning marketplace. For instance, could this valuation approach inadvertently favor certain types of models or tasks over others, or influence the development and use of machine learning models in ways that could have unintended consequences? They could also consider the potential impacts on data privacy, given that model valuation, unlike data valuation, does not require centralization of potentially private data.

---

> ### Author Rebuttal · Authors · 2023-08-08
>
> We thank Reviewer RTGV for taking the time to review our paper and providing very detailed feedback and comments, especially saying that our work presents "a novel approach", "is quite innovate" and "a valuable addition to the machine learning and statistical literature". We wish to address the feedback and questions as follows.
>
> W1.
> > ... It should address situations where the available data is sparse or not balanced across classes. A detailed discussion or a possible solution ... .
>
> __We address a highly imbalanced case in our experiments, i.e., KDD99 dataset, using the class-specific Dirichlet abstractions weighted by the size of each class-specific query set__ (lines 345-348). Our results show that this approach is _effective and necessary_. Figure 4 (left) under the non-class-specific approach is unable to distinguish models trained on different amounts of training data (with 100-fold difference in size) while Figure 4 (right) under the class-specific approach is able to do so. We will make this discussion more explicit in our revision.
>
> W2.
> > ... there is a lack of clear guidelines or a framework for determining the optimal trade-off. ... Adding a more formal discussion or a proposed methodology to address this trade-off would be beneficial.
>
> __Adopting the highest level of abstraction can already be quite effective__ (shown in our experiments for MNIST, CIFAR-10, and two real-world datasets MedNIST and DrugRe) if the query set is _not highly imbalanced_.
> For a highly class-imbalanced query set (such as KDD99 in our experiments), __adopting the class-specific Dirichlet abstractions weighted by the size of the class-specific query set is effective__ (lines 345-348). We will include this in our revision and believe that a more extensive and formal framework is a very useful future direction.
>
> W3.
>
> > ... assumes homogeneity across models after the Dirichlet abstraction, ...
>
> __All models (regardless of architectures) for the same learning task__ (i.e., same feature and label spaces) __can be represented via their corresponding Dirichlet abstractions, which are homogeneous by design, not assumption__. As an example, the Dirichlet abstractions of a logistic regession and a CNN are both Dirichlet distributions, but with possibly different parameters.
>
> > Some models might have specific characteristics that cannot be captured by the Dirichlet distribution.
>
> [2] provides a polynomial sample complexity of the query set w.r.t. the error and a model's actual predictive distribution. It is an interesting future direction to study its implications for model valuation.
>
>
> W4.
> > ... a comparison with other existing model valuation methods
>
> __There are limited existing methods applicable for comparison__: The closest work is [55], but it is restricted to the binary classification, thus not applicable to $C$-way classification (in lines 230-231 and 370-371). We include an additional discussion on other works from an economics viewpoint (without a machine learning focus) in App. B.2 (lines 844-851).
>
> Hence, __we investigate some intuitive baselines where our proposed approach produces consistent results to these intuitive baselines__ (i.e., predictive accuracy in Figure 3, F$1$ score in Figure 4 right, training data sizes in Figure 4 left). A further discussion is included in App. B.2 (lines 833-843). Moreover, we have also __derived theoretical connections between our approach to some sophisticated criteria such as fairness, and robustness__ (Sec. 3.2 and Table 1).
>
> Q1.
> > How does the method perform when the available data is sparse or unbalanced across classes?
>
> See response for W1.
>
> > ... how their approach handles diverse model architectures, especially those with characteristics that cannot be adequately captured by a Dirichlet distribution?
>
> See response for W3.
>
> > ... how to choose an optimal trade-off between the level of Dirichlet abstraction and the size of the query set?
>
> See response for W2.
>
> > Some of the assumptions ... (like model homogeneity post-abstraction) might not hold ... . Could the authors elaborate on the implications if these assumptions are violated and how such situations could be handled?
>
> - On "model homogeneity", see response for W3.
>
> - A key assumption (for Theorem 1) is fusion-inreases-similarity (lines 272-273). __We specifically derive a sufficient condition for identifying when this assumption holds__ (Prop. 4 in App. A.3) and provide an interpretation.
>
> L1.
> > ... the sensitivity of the model's valuation ..., and the limitations of the Dirichlet abstraction in approximating a model's predictive pattern or distribution.
>
> We thank the reviewer for the suggestion and will incorporate this in our revision.
>
> L2.
> > ... could this valuation approach inadvertently favor certain types of models or tasks over others, or influence the development and use of machine learning models in ways that could have unintended consequences?
>
> __This is an important motivation to consider a diverse range of evaluation criteria, from the more intuitive predictive accuracy to the more sophisticated ones such as fairness and robustness__. We are happy to provide further elaboration on this, due to the character limit of the rebuttal.
>
> L3.
> > They could also consider the potential impacts on data privacy, given that model valuation, unlike data valuation, does not require centralization of potentially private data.
>
> The __privacy regulation on data is an important motivation of our work__ (lines 26-30) and is further discussed in App. B. 1.
>
> We thank the reviewer for the detailed feedback and for the suggestions, and will incorporate these discussions in our revision. We hope our response clarified the questions and has helped improve your opinion of our work. We are happy to provide further clarifications or elaboration.

---

> ### Author Response · Authors · 2023-08-21
> **Gentle reminder for response**
>
> We wish to thank Reviewer RTGV for the positive feedback ("novel approach", "quality of research seems to be high") and the questions, and are keen on finding out whether our response has clarified your questions, since the discussion period is coming to an end soon (in less than 15 hours). We really appreciate your feedback and acknowledgement. Thank you.

---

### Official Review · Reviewer_tCDm · 2023-07-13

**Soundness:** 3 good
**Presentation:** 3 good
**Contribution:** 3 good
**Rating:** 7
**Confidence:** 4

**Summary:**

The paper considers the problem of assigning a value to a ML model (e.g., in a marketplace with multiple models). The proposed idea is to estimate (via MLE) the Dirichlet abstraction of a model (potentially conditioned on the class) and to compute the Shapley value of a game where the value function is the Hellinger distance between the Dirichlet abstraction of a model that fuses the models in the coalition and an (almost) optimal model. The paper shows that this proposal is well-behaved and allows for learning the Shapley value to reduce the complexity (the number of Shapley values to compute). The paper also shows a number of experiments on standard datasets (MNIST, CIFAR et al.) and compares to other more standard quality metrics.

**Strengths:**

The problem of valuing a model with black-box access (i.e., we can access predictions but not the model itself) is interesting because of many considerations that restrict sharing data and model parameters. The paper proposes an original solution in considering the Dirichlet abstraction of the model to compute metrics. This indeed helps because then models fusion as well as Hellinger distance can be computed easily. In a sense, the framework itself is the main contribution more than the theoretical results (which is totally fine).

One of the advantages of the Dirichlet abstraction framework is that it handles easily and naturally multiple classes, as opposed to binary classification which is much more standard in that type of literature.

I find the paper well written and easy to follow (up to the issue of Figures to understand numerical experiments, which I discuss later).

**Weaknesses:**

I had a doubt about the Shapley value: Shapley assumes an axiom of efficiency. Here, $Z$ seems free. So $\phi_i$ will be the Shapley value only for a particular value of $Z$. Does it have any impact?
More generally, this should be clarified. For instance, Thm 1 clearly depends on $Z$ and I have not seen that said anywhere. In the proof, it looks like $Z$ is taken to be $1$, is that the case? Does it correspond to Shapley then?

The condition in Prop 4 (App A), which implies that fusion increases similarity, is not too explicit. Is there a nice intuitive interpretation of it? And does it hold (theoretically) for the models used in the experiments for instance?

The notation can be improved in some places. For instance, l. 208, $n$ is the size of a subset that depends on $\mathcal{C}$ whereas $n$ is almost always the size of the full set $N$. Also the fact that $C$ is the dimension of the label whereas $\mathcal{C}$ is a subset of $N$ is confusing.

The definition of $\mathbb{Q}^*$ poses some questions. It is in a sense supposed to represent an optimal model, but then some noise is added with range $[0, 0.01]$ to avoid degeneracy in MLE. So it means that theoretically, there could be a better model with a non-zero distance and hence strictly speaking the Hellinger distance is not decreasing with the model quality. Is it correct? If so, isn't there a nicer way to define $\mathbb{Q}^*$ theoretically?

On the learning of the Shapley value: the proposal is to compute the Shapley value for a subset of models (say, 20 out the 150 models) and to use that to infer the value of others because the Shapley value has some regularity wrt the Dirichlet abstraction. This is nice, but poses two questions:
a- is it obvious that one can just compute the Shapley values of a subset of models? Normally, computing the Shapley value requires computing the marginal increment to some subcoalitions (even if not all, using MC). Here, we cannot do it for coalitions that aren't in the subset of 20, so how do we do? This requires clarification. Perhaps the special form of the game allows that but it needs to be clarified.
b- in the experiments (l. 305-312), my understanding is that the paper computes the Shapley value of all 150 models (used as ground-truth), and then uses only a subset of them to infer the others. That by-passes the issue I mentioned just above... but then it does not allow to get the reduced complexity in practice so the issue stands completely. I think a better experiment would be to compute the Shapley value of the 20 models using whatever solution the authors came up with to my point a- above (which needs to be clarified), and then using it to infer the SV of the others, comparing it to the case where one can indeed compute all 150 SVs.

Figures are really too small. I understand the page limit but it is excessive. Some of them are literally unreadable (like the plot is hidden behind the legend, e.g., Fig 3 left, for MLP). Also in many of them, I could not understand what the x-axis is (mostly because it was not written).


UPDATE AFTER REBUTTAL: I raise my score to 7, due to a better understanding, in particular of the experiments.

**Questions:**

[Please also see the weaknesses part where I put questions as well (probably the most important); I am putting here whatever I have not already written above.]

- The Dirichlet abstraction allows handling multiple classes nicely, but it was not clear to me what would simplify or get more standard if we have binary classification.

- I understand why it helps (l. 210-220) with the Shapley computation, but it would be good to better explain why doing the fusion as it is done makes sense from the prediction perspective.

- For clarification: the Dirichlet abstraction (e.g., l. 100) is not allowed to depend on x?

- Is it possible to extend the framework to assign different weights to different types of errors?

- It may be good to add a bit more background on the Dirichlet distribution and Hellinger distance (in App), just enough to help follow the rest of the paper for those who are not familiar with these.

- Fig 1: although I understand it doesn't matter for the point the paper makes there, I find that plots 1-2 aren't similar to plots 3-4. Is this normal?

**Limitations:**

See above.

---

> ### Author Rebuttal · Authors · 2023-08-08
>
> We thank Reviewer tCDm for taking the time to review and providing such detailed feedback and comments, and for finding our paper "interesting" and "well written". We wish to address the feedback and questions as follows.
>
> W1.
> > So $\phi\_i$ will be the Shapley value only for a particular value of $Z$. Does it have any impact? More generally, this should be clarified. ... Thm 1 clearly depends on $Z$ and I have not seen that said anywhere.
>
> __Theorem 1 holds regardless of the value of $Z$__, by identifying the suitable constant $L$ in Lemma 6 in App. A.4, though the bound can become looser for a larger $Z$ (due to a large $L$ in Lemma 6).
>
> > ... it looks like $Z$ is taken to be $1$ , is that the case? Does it correspond to Shapley then?
>
> W.l.o.g., we set $Z=1$ and it does recover the original Shapley value.
>
> W2.
> > The condition in Prop 4 (App A), ... Is there a nice intuitive interpretation of it?
>
> __If the "shapes" of $\mathcal{Q}\_i$ and $\mathcal{Q}\_{i'}$ are very different, then fusing each to a common $\mathcal{Q}\_{\mathcal{C}}$ increases the resulting similarity as it "evens out" their difference (lines 732-736).__ This is further elaborated in the lines 724-731, 732-736 of App. A.3.
>
> > And does it hold (theoretically) ... ?
>
> Our empirical verification of Theorem 1 (which requires Prop. 4) observes that the the models which are similar do lead to similar MSVs in lines 278-299 and Figure 2 and Table 2. Theoretical verifictaion is an interesting future exploration.
>
> W3.
> > notation can be improved in some places
>
> We clarify that $N$ denotes an integer, while $[N]$ is the set $\{1,\ldots, N\}$. We thank the reviewer for raising this and will revise our notations.
>
>
> W4.
> > there could be a better model ... Is it correct? If so, isn't there a nicer way to define theoretically $\mathcal{Q}^\*$?
>
> We first highlight that __our theoretical results do _not_ require this definition of $\mathcal{Q}^\*$__. Then,
> though there is a possibly better hypothetical model with a non-zero distance, empirically we find our definition effective, possibly because __such hypothetical models are very rare__. Nevertheless, exploring alternative definitions of $\mathcal{Q}^\*$ is an interesting direction.
>
> W5.
> > a- is it obvious that one can just compute the Shapley values of a subset of models?
>
> __We are considering a game of $150$ models to only obtain the MSVs of $20$ models__, instead of a game of only $20$ models. This elaborated further in App. C.2 (lines 881-888).
>
> > b- ... my understanding is that the paper computes the Shapley value of all 150 models (used as ground-truth), and then uses only a subset of them to infer the others.
>
> This is the correct understanding of the learning approach, which only sees a subset of the ground-truths as the training data. The __remaining unseen subset is used only for evaluation__.
>
> > a better experiment would be to compute the Shapley value of the 20 models ... compute all 150 SVs.
>
> __This is indeed our expeirment setting__. Table 3 shows comparison between the predicted MSVs and the obtained ground-truths to have low prediction errors, the saving in computational cost is from __only needing to obtain MSVs for a subset of models__, further elaborated in App. C.4 (lines 869-880).
>
> W6.
> > I could not understand what the x-axis is
>
> We will increase our figure size in our revision, and clarify here: the x-axis (and y-axis) in Fig.2 is the model indices $i$; the x-axis Fig. 3 (left) is the model types, the x-axis Fig. 3 (right) is the predictive certainty (see lines 337-339); the x-axis in Fig. 4 is the size of training data (see its caption, and lines 340-341).
>
> Q1.
> > but it was not clear to me what would simplify or get more standard if we have binary classification.
>
> Theoretically, $C>2$ or $C=2$ does _not_ make a significant difference, as the intended design. In contrast, [55] have specifically exploited the simpler structure of $C=2$ and thus it seems difficult to extend their method to $C>2$. For practice, a smaller $C$ means fewer class-specific query sets to collect (lines 357-358).
>
> Q2.
> > ... why doing the fusion as it is done makes sense from the prediction perspective.
>
> We wish to highligh that the __fused Dirichlet abstraction $\mathcal{Q}\_{\mathcal{C}}$ is _not_ a predictive model itself__ (i.e., it can not be used to produce predictions by taking as input queries), and it is __primarily designed to be amenable to valuation.__
>
> Q3.
> > ... the Dirichlet abstraction (e.g., l. 100) is not allowed to depend on $x$?
>
> **A Dirichlet abstraction is a Dirichlet distribution, so it does not depend on $x$**. For a model $\mathbf{M}\_i: \mathcal{X} \mapsto \triangle(C)$, for a random variable $X \sim P_X$ whose $\text{supp}(X) = \mathcal{X}$ (and $P_X$ represents the task), $\mathbf{M}\_i(X)$ is a distribution over $\triangle(C)$, and this is represented using the Dirichlet abstraction (lines 101-102).
>
>
> Q4.
> >  ... extend the framework to assign different weights to different types of errors?
>
> We consider the "types of errors" to be different model evaluation criteria, and a user __can easily linearly combine different model evaluation criteria__ (Sec. 3.2 and Table 1), by exploiting P3 (linearity) and implementing the selected criterion from Table 1.
>
> Q5.
> > ... a bit more background on the Dirichlet distribution and Hellinger distance.
>
> App. A.2 includes this background on Dirichlet distribution and Hellinger distance, and we will incorporate the suggestion in our revision.
>
>
> Q6.
> > Fig 1: ... I find that plots 1-2 aren't similar to plots 3-4. Is this normal?
>
> Yes, plots 1-2 show the probability vectors while plots 3-4 show the densities of the learnt Dirichlet abstractions from these
> probability vectors.
>
> We thank Reviewer tCDm for the detailed review and questions, and hope our response has helped clarified the questions and  raised your opinion of our work. We are happy to provide further elaboration in discussion (due to character limit of rebuttal).

---

> > ### Comment · Reviewer_tCDm · 2023-08-12
> >
> > I thank the authors for their response. It helps understand the paper better. It still find it a pity that we do not have a way to theoretically show that the models used satisfy the assumption of Prop 4, but ok. Overall, I find this to be an interesting contribution and will raise my score to 7. That said, I do see the critics made by other reviewers. I do not disagree with them, I simply feel that the paper is acceptable despite those---but of course this is only one option amongst several.

---

> > > ### Author Response · Authors · 2023-08-15
> > > **Thank the reviewer for the positive feedback**
> > >
> > > We thank Reviewer tCDm for the quick response and in particular for appreciating our contribution and raising the score.

---

### Author Rebuttal · Authors · 2023-08-10

We thank all the reviewers for taking the time to review our paper and providng the detailed comments and positive feedback:

- Our studied problem is interesting and well-motivated (Reviewers tCDm, bude & Coej);

- Our approach is novel and creative, and the quality of our research is high (Reviewer RTGV); our solution is solid and well-thought-out, and our theoretical statements are sound (Reviewer bude); our approach using the Shapley value is well-motivated (Reviewer Coej);

- Our paper is well structured and written (Reviewers tCDm & bude);

- Our paper presents a valuable addition to the machine learning and statistical literature (Reviewer RTGV).

In our prepared response to your feedback and questions, with main points summarized below, we have
- Provided additional clarifications on some theoretical results:

    - On the effect of $Z$ in Equ.(2), condition in Prop. 4, definition of $\mathcal{Q}^*$ and our proposed learning approach (Reviewer tCDm);
    - On the homogeneity of Dirichlet abstractions and other assumptions (Reviewer RTGV);
    - On the implications of P4 and P3 (Reviewer bude);
    - On the formal problem setting, theoretical motivation and justification of the proposed Dirichlet abstraction, to highlight our main contributions (Reviewer Coej).

- Provided discussion from a practical viewpoint:

    - On the application to the simpler binary classification, and extension to combining different errors (Reviewer tCDm);
    - On the preparation of the query set (Reviewers RTGV & bude);
    - On the trade-off between query set size and abstraction level, and dealing with an unbalanced query set with very sparse data for some classes (Reviewer RTGV);
    - On the bounds of query set size for potential reverse-enginnering (Reviewer bude)  .

We thank all the reviewers for reviewing our paper and their reviews, and hope that our response has clarified your questions and helped raise your opinions of our work. We are happy to provide further clarifications during the discussion period.

---

### Decision · Program_Chairs · 2023-09-21

**Decision:**

Accept (poster)

**Comment:**

All of the reviewers agree that the paper proposes an interesting well-motivated problem and offers a good initial solution that may be of interest to the community. The reviewers also found the Dirichlet abstraction proposed in the paper a novel and effective contribution. However, the reviewers identified several opportunities to improve the paper. The authors clarified several questions around the theoretical results, and discussed the practicality of the method, which should be added to the paper to improve its exposition. One of the reviewers astutely pointed out a subtle distinction between black-box access (claimed in the paper) and grey-box access (implicitly required to construct common query sets for a task). These nuances are important to clarify, even in a first paper on a proposed problem formulation.